# A rapid and sensitive assay for quantifying the activity of both aerobic and anaerobic ribonucleotide reductases acting upon any or all substrates

**Talya S. Levitz**[1], **Gisele A. Andree**[2], **Rohan Jonnalagadda**[1], **Christopher D. Dawson**[1], **Rebekah E. Bjork**[3], **Catherine L. Drennan**[1,2,3,4] *

**1** Department of Biology, Massachusetts Institute of Technology, Cambridge, MA, United States of America, **2** Department of Chemistry, Massachusetts Institute of Technology, Cambridge, MA, United States of America, **3** Howard Hughes Medical Institute, Massachusetts Institute of Technology, Cambridge, MA, United States of America, **4** Center for Environmental Health Sciences, Massachusetts Institute of Technology, Cambridge, MA, United States of America

* cdrennan@mit.edu

**Data Availability Statement:** All relevant data are within the manuscript and its Supporting Information files.

## Abstract

Ribonucleotide reductases (RNRs) use radical-based chemistry to catalyze the conversion of all four ribonucleotides to deoxyribonucleotides. The ubiquitous nature of RNRs necessitates multiple RNR classes that differ from each other in terms of the phosphorylation state of the ribonucleotide substrates, oxygen tolerance, and the nature of both the metallocofactor employed and the reducing systems. Although these differences allow RNRs to produce deoxyribonucleotides needed for DNA biosynthesis under a wide range of environmental conditions, they also present a challenge for establishment of a universal activity assay. Additionally, many current RNR assays are limited in that they only follow the conversion of one ribonucleotide substrate at a time, but in the cell, all four ribonucleotides are actively being converted into deoxyribonucleotide products as dictated by the cellular concentrations of allosteric specificity effectors. Here, we present a liquid chromatography with tandem mass spectrometry (LC-MS/MS)-based assay that can determine the activity of both aerobic and anaerobic RNRs on any combination of substrates using any combination of allosteric effectors. We demonstrate that this assay generates activity data similar to past published results with the canonical *Escherichia coli* aerobic class Ia RNR. We also show that this assay can be used for an anaerobic class III RNR that employs formate as the reductant, i.e. *Streptococcus thermophilus* RNR. We further show that this class III RNR is allosterically regulated by dATP and ATP. Lastly, we present activity data for the simultaneous reduction of all four ribonucleotide substrates by the *E. coli* class Ia RNR under various combinations of allosteric specificity effectors. This validated LC-MS/MS assay is higher throughput and more versatile than the historically established radioactive activity and coupled RNR activity assays as well as a number of the published HPLC-based assays. The presented assay will allow for the study of a wide range of RNR enzymes under a wide range of conditions, facilitating the study of previously uncharacterized RNRs.

**Funding:** This work was supported by NIH grant GM126982 to C.L.D and NSF GRFP 2017246757 to T.S.L. as well as NIH T32 Training Grant 5T32GM007287 awarded to the MIT Biology Department. C.L.D is a Howard Hughes Medical Institute Investigator and Senior Fellow for the Canadian Institute for Advanced Research Bio-Inspired Solar Energy program. Support for all LC-MS/MS data collection and analysis was provided to the MIT Center for Environmental Health Sciences (CEHS) by a core center grant P30-ES002109 from the National Institute of Environmental Health Sciences, National Institutes of Health. Support for the thermal denaturation assay was provided to the MIT BioMicro Center by the Koch Institute Support (core) Grant P30-CA14051 from the National Cancer Institute. The funders had no role in the study design, data collection and analysis, decision to publish, or preparation of the manuscript.

**Competing interests:** No. The authors have declared that no competing interests exist.

## Introduction

Ribonucleotide reductases (RNRs) use radical-based chemistry to convert ribonucleotides to deoxyribonucleotides, maintaining the cellular deoxyribonucleotide pools that are needed for DNA biosynthesis and repair (Fig 1). RNRs are medically important enzymes; human RNR is a drug target for cancer, and bacterial RNRs have been proposed as antibiotic targets [1–6]. The ribonucleotide reductase enzyme family is broken broadly into three main classes, named class I, class II, and class III, based upon the cofactor used to generate a catalytically-essential thiyl radical species in the enzyme active site (Fig 1; Table 1) [2, 7, 8].

Class Ia enzymes, whose membership includes both the canonical *E. coli* class Ia RNR (*Ec*RNR) and human RNR, are obligately aerobic enzymes that use a di-iron tyrosyl radical cofactor, housed in a β2 subunit, to generate the thiyl radical species in the active site of the catalytic (α2) subunit (Fig 1; Table 1) [2, 9, 10]. Despite having been studied for the past sixty years, new subclasses of class I RNRs are still being discovered, demonstrating the diversity and unknown potential even within classes of RNRs [11–14]. Class II RNRs are oxygen-independent enzymes that use 5′-adenosylcobalamin to generate the catalytically-essential thiyl radical species [15, 16]. In contrast to class I RNRs, which only act on ribonucleoside diphosphate (NDP) substrates, class II RNRs are divided in whether they act on ribonucleoside triphosphate (NTP) substrates or NDP substrates [17–24]. Lastly, the class III RNRs are obligately anaerobic enzymes due to the presence of a post-translationally installed glycyl radical cofactor that is oxygen-sensitive [25–27]. Unlike class I and class II enzymes that typically use a reducing system comprised of thioredoxin (Trx), thioredoxin reductase (TrxR) and nicotinamide adenine dinucleotide phosphate (NADPH), most characterized class III RNRs (NrdDs) are of the NrdD1 subclass, which use formate as the reductant, although there are two other NrdD subclasses that use different reductants (Table 1) [2, 8, 28].

RNRs are tightly regulated in order to keep cellular ribonucleotide and deoxyribonucleotide pools balanced; disruption of these nucleotide pools via disruption of RNR activity or RNR regulation can have disastrous consequences for the cell [29–33]. In addition to the substrate binding site, each RNR catalytic subunit contains either one or two allosteric nucleotide binding sites (Fig 1). The first site, present in all RNRs, is known as the specificity site and structurally informs which of the four ribonucleotide substrates is preferred in the active site [7, 9, 34, 35]. Although all classes of RNRs contain a specificity site, how the nucleotide in the specificity site determines what nucleotide is bound in the active site has not been fully investigated outside of the prototypical class Ia *Ec*RNR and a few other RNRs [34–39]. The second site, termed the activity site, is only present in RNRs containing a conserved N-terminal cone domain [40, 41]. In RNRs containing this cone domain, the activity site acts as an on/off switch for the enzyme: binding dATP inactivates the enzyme, whereas displacement of dATP by ATP reactivates the enzyme [2, 41–43]. Approximately 47% of class I RNRs and 7% of class II RNRs contain a cone domain, although not all RNRs that contain a cone domain have activity regulation and at least one RNR without a full cone domain has exhibited activity regulation [44–47]. Unfortunately, although 76% of the known class III RNRs contain one or multiple cone domains, neither of the two published structures of class III RNRs are RNRs that have a cone domain [34, 46, 48]. Much is still unknown about the structural and biochemical basis of activity regulation, if present, in RNRs that are not class Ia RNRs.

The development of a universal RNR activity assay has been a longtime goal that has been complicated by all of the factors mentioned above: two different substrate phosphorylation states, differences in reducing systems used, presence/absence of allosteric activity regulation, complex allosteric specificity relationships, and oxygen requirements among and between the class I, class II, and class III RNRs (Table 1). The main activity assay that has historically been

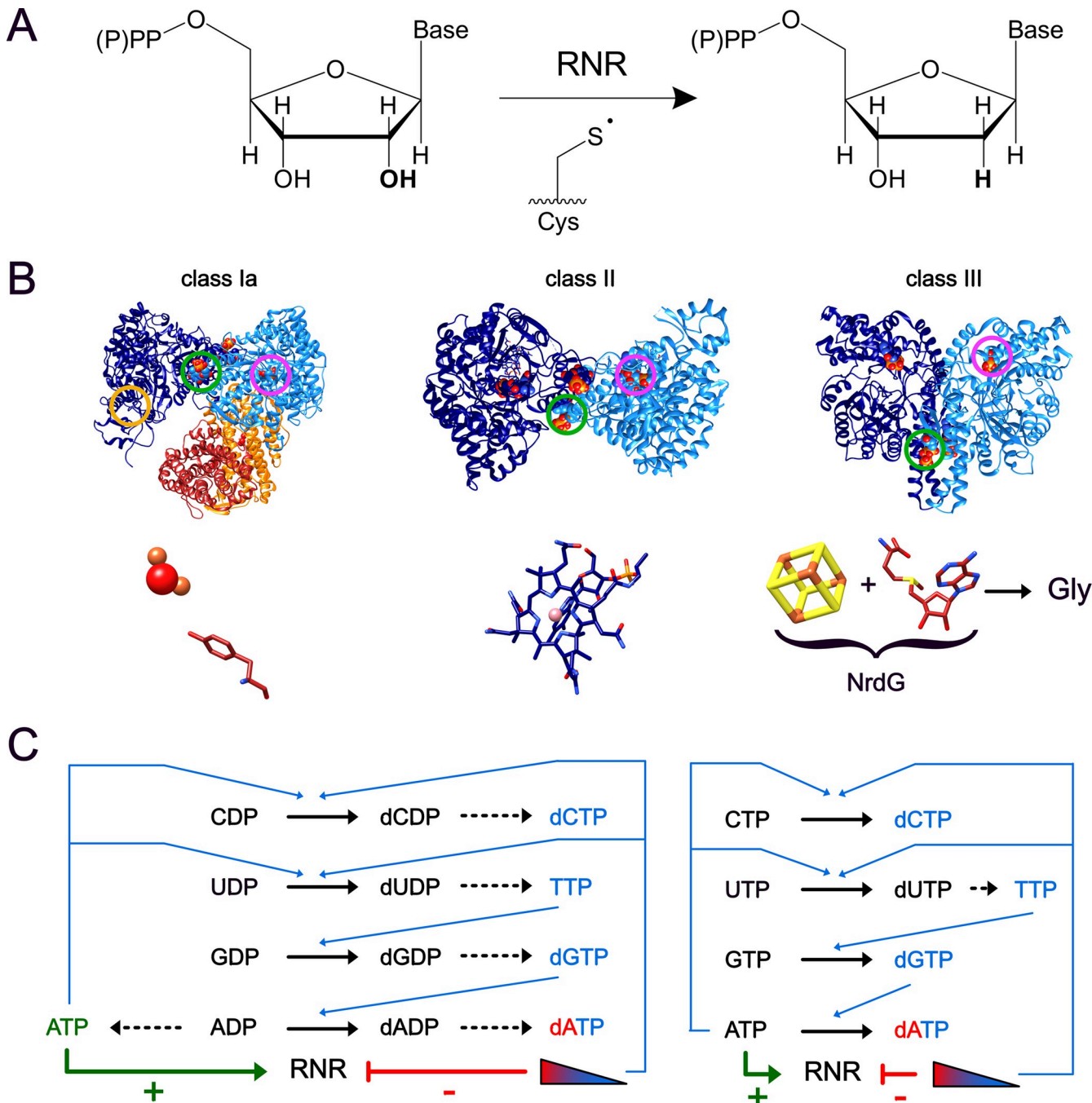

**Fig 1. An overview of the ribonucleotide reductase reaction, classes, and allosteric regulation. (A)** Ribonucleotide reductase catalyzes the conversion of ribonucleotide di- or triphosphates to their corresponding deoxyribonucleotide via a catalytic cysteine radical. **(B)** Examples of class I, class II, and class III enzymes with their corresponding metallocofactors and radical harbors. In each structure, one specificity effector site is circled in green and one substrate site is circled in magenta. The *E. coli* class Ia RNR (left) is shown in the active form with the catalytic subunits (NrdA) shown in blue and the radical-generating subunits (NrdB) shown in orange/red (PDB 6W4X) [52]. Bound nucleotides (GDP substrate and TTP specificity effector) are shown as spheres. The activity effector site, which does not have any bound nucleotide in this structure, is circled in orange. The diiron-oxo center and tyrosyl radical shown below the structure are found in the radical-generating subunit and are specific to class Ia RNRs; see Table 1 for the radical harbors of other subclasses of class I RNRs. The *Thermotoga maritima* class II RNR (NrdJ; middle) is shown with the bound nucleotides (GDP substrate and TTP specificity effector) shown as spheres (PDB 3O0O) [34]. The *S*-adenosylcobalamin (B$_{12}$) cofactor found in each subunit is shown as sticks and is highlighted below the structure. The *Escherichia coli* T4 bacteriophage class III RNR (NrdD; right) is shown with bound nucleotides (dGTP) shown as spheres (PDB 1HK8) [85]. The activase NrdG is structurally uncharacterized and contains the 4Fe-4S cluster shown below the structure. NrdG in complex with *S*-adenosylmethionine generates a glycyl radical is generated on the catalytic subunit via a transient interaction. **(C)** Allosteric regulation of RNRs that use NDPs (left) or NTPs (right) as substrates. Following

reduction of nucleotides, the deoxyribonucleotide products are either first converted to dNTPs (left) or directly used (right) as specificity effectors for determining which nucleotides bind the active site for future reduction reactions (blue arrows). In some but not all RNRs, ATP acts as an overall positive activity effector and dATP at high concentrations acts as an overall negative activity effector; ATP can also act as a specificity effector in vitro and possibility in vivo. Figure modified from [78].

used to measure RNR activity is a radioactivity-based assay [10, 49–51]. In this assay, RNR converts radioactive substrate (generally tritiated CDP or CTP) to radioactive product (tritiated dCDP or dCTP). At each time point in the reaction, the reaction is stopped either chemically or via heat before product is separated from the remaining substrate through the use of a boronate affinity column that selectively binds the 2′ and 3′ hydroxyl groups of the substrate's ribose. The flow-through from the column, which contains only radioactive deoxyribonucleotide product, is then quantified for radioactivity using a liquid scintillation counter. Although this radioactive assay is well established in the field, it has numerous drawbacks, including generally only being performed on CDP as opposed to other substrates due to the need for HPLC separation for GDP reduction and expense/availability of radiolabeled UDP and ADP [52]. Additionally, the radioactive assay can only measure reduction of a single ribonucleotide species at one time, despite the enzyme reducing all four ribonucleotide substrates simultaneously in vivo. The use of radioactive substrates also requires special training, dedicated space, and radioactivity licenses, and can be particularly challenging for class III RNRs, which must be assayed in anaerobic chambers that are often not certified for radioactivity.

A second historically-utilized assay in the RNR field is a coupled assay that measures the spectrophotometric change when NADPH is oxidized to NADP$^+$ [53, 54]. This assay does not directly measure products, and so it only works for RNRs that use an NADPH/Trx /TrxR reducing system, notably excluding the class III RNRs that use formate as the reductant. Additionally, the assay requires the cloning and purification of the Trx and TrxR proteins that correspond to each ribonucleotide reductase being measured; these enzymes may be difficult to procure or express, such as in the example of the human TrxR, which contains a selenocysteine residue [55, 56]. The coupled assay often displays large variability between replicates and, similarly to the radioactivity assay, can only be used to measure reduction of one single substrate/effector pair at a time.

In light of the above concerns for the most historically-established assays, numerous groups have turned to HPLC-based assays to separate and detect effectors, substrates, and products. As early as the 1980s, an HPLC protocol was published that separated substrate from product but required radioactivity to detect product traces [52, 57]. Subsequent protocols utilizing

**Table 1. Comparison of class I, class II, and class III RNRs highlighting the differences that make a universal activity assay challenging.** Subclasses of class III RNRs are listed as NrdD1, NrdD2, and NrdD3.

| | Class I | Class II | Class III |
|---|---|---|---|
| **Oxygen requirement** | Aerobic | Oxygen-independent | Anaerobic |
| **Subunits Required for Activity** | Catalytic subunit and radical-storage subunit | Single subunit | Catalytic subunit and activase (activase required for installation of glycyl radical species G•) |
| **Radical harbor** | Tyr• (class Ia and Ib) Unpaired electron in metal center (class Ic and Id) DOPA• in presence of NrdI (class Ie) | 5'-adenosylcobalamin (B$_{12}$) | S-adenosyl methionine/[4Fe-4S]/ G• |
| **Substrate** | NDPs | NDPs or NTPs | NTPs |
| **Reducing system** | Thioredoxin, thioredoxin reductase, and NADPH Or Glutaredoxin, glutathione, glutathione reductase, and NADPH | Thioredoxin, thioredoxin reductase, and NADPH | Formate (NrdD1) Thioredoxin (NrdD2) Glutaredoxin-like NrdH (NrdD3) |

HPLC-based detection of products have been generated that do not need radioactivity, greatly increasing the access of researchers to measure product formation of various RNRs [11, 21, 22, 44, 58–63]. These assays have been effectively used to measure the absolute or relative activities of a wide variety of RNRs from numerous classes, including in one case under anaerobic conditions [62] and, in multiple other cases, measuring the reduction of all four substrates simultaneously [21, 59, 60]. However, many of these assays are still hampered by long run times (40–180 minutes) and/or relatively high flow rates (1–1.5 mL/min), which makes the analysis of many samples a time- and reagent-consuming process [44, 58, 60, 61]. Additionally, in order to get enough product to see each peak on the HPLC trace, some assays are completed over a longer period of time (15–60 mins) [11, 21, 22, 44, 58–61, 63], and assay speed is not just a matter of convenience: RNRs have radical cofactors with various degrees of stability necessitating an assay speed faster than radical cofactor decay. Part of the reason that the class Ia *Ec*RNR is the workhorse of the RNR family is that it has a tyrosyl radical that is relatively stable (lifetime of several days) [64]. To study RNRs with less stable radical cofactors (e.g., the half-life of the Y• of the mouse β2 is 10 minutes), assay speed is an important consideration [65]. Lastly, due to the focus on the biological questions at hand and the study of RNRs from diverse organisms in the above references, there is no published method validation or comparisons of activities generated using the HPLC-based assays to established values in the field.

With the many assays present in the literature in mind, we set out to generate and validate an assay that could measure reduction of all four ribonucleotides simultaneously from any class of RNR. An LC-MS/MS-based assay was chosen for its ability to easily differentiate and directly measure the four deoxyribonucleotide products from a mixture of substrate and products. Here, we describe an assay that (1) uses unlabeled substrates, a low flow rate, and short run times; (2) differentiates all four deoxyribonucleotide products from one another; and (3) can be done either in an anaerobic chamber or on the benchtop, making the assaying of class III RNRs more amenable.

## Results

### An LC-MS/MS-based assay can determine the activity of ribonucleotide reductase enzymes

We developed an assay that can be used to measure the activity of both aerobic and anaerobic ribonucleotide reductase enzymes (Fig 2). In the assay (final volume 240–250 μL), mastermixes consisting of the α2 subunit of *Ec*RNR (NrdA; final concentration 0.1 or 0.5 μM), nucleotides (variable concentrations), Trx (final concentration 30 μM), TrxR (final concentration 0.5 μM), and NADPH (final concentration 200 μM) or of activated class III RNR from *Streptococcus thermophilus* (*St*NrdD; final concentration 0.1 μM), nucleotides (variable concentrations) and formate (final concentration 12.5 mM) are generated and allowed to equilibrate at 37˚C. For all assays described below, the RNR subunit being assayed was held as the limiting reagent at 0.1 μM with nucleotide concentrations varied based on experimental question; optimization of concentration will be necessary for RNRs not presented here to determine what concentrations yield measurable activity within the linear range of the assay.

A zero point is taken prior to initiation, and then the remaining mastermix is initiated with a small amount (generally 1–5 μL, concentration-dependent) of either β2 (NrdB for *Ec*RNR; final concentration 0.5 or 0.1 μM) for class Ia RNRs or nucleotides (variable concentrations) for class III RNRs and mixed via pipetting while being held at 37˚C. The reaction proceeds at 37˚C with time points taken every 30 seconds for 120 seconds. At each time point, a 40 μL aliquot of the reaction is inactivated by rapid heating to 95˚C. We confirmed that the thermophilic *St*NrdD is completely inactivated at 95˚C; the melting temperature of the enzyme was

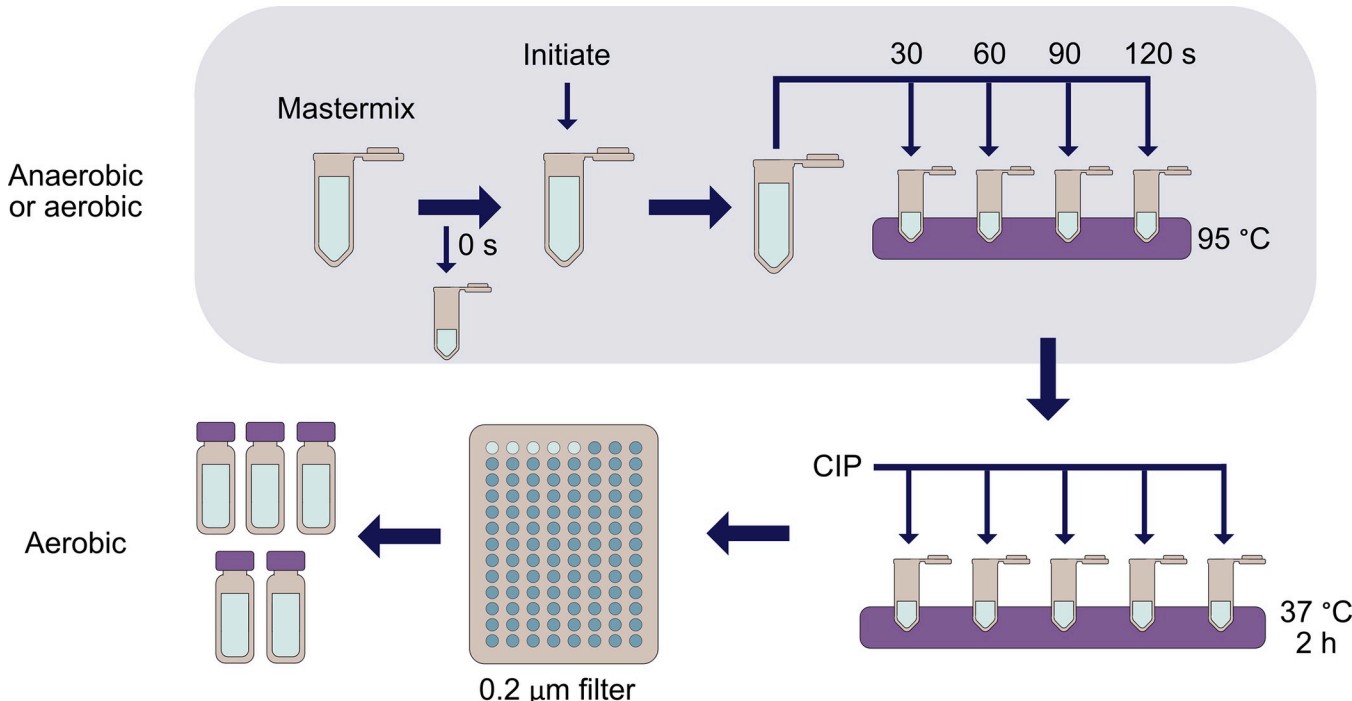

**Fig 2. Overall scheme of LC-MS/MS RNR activity assay.** Either aerobically or anaerobically, a mastermix is created and a zero point is taken prior to initiation with beta (aerobic assays) or nucleotides (anaerobic assays). The reaction is allowed to proceed for 120 s with time points taken every 30 s. Inactivation of the reaction occurs by heating the sample to 95°C in a thermocycler or heat block. After inactivation, anaerobic assays are removed from the box and all assays have 1 μL of calf intestinal phosphatase (CIP) added to dephosphorylate the nucleotides at 37°C for 2 hours. The samples are then filtered through a 0.2 μm filter and transferred to mass spectrometry vials for analysis via LC-MS/MS.

determined to be approximately 50°C in the presence of nucleotides, which is reached in under 5 seconds after transferring to the 95°C thermocycler (S1 Fig, S1 Table). Following the heat step, all nucleotides are dephosphorylated with the addition of calf intestinal phosphatase (CIP) prior to filtration and transfer to mass spectrometry vials. The CIP dephosphorylation step was optimized (S2 Fig).

Activity is determined by extracting multiple reaction monitoring (MRM) curves for each product from the overall total ion chromatogram (TIC) for each LC-MS/MS run (Fig 3). MRM mode on the mass spectrometer filters the total incoming ions, which are used to generate the TIC, by the individual parent and fragmented ion masses for each product, allowing for the generation of a separate curve for each product and deconvolution of overlapping peaks in the TIC, and ensuring the measured signal is due solely to the product of interest. Slight run-to-run variability in retention time (on the order of maximum ± 0.3 mins) is occasionally seen for some products, but retention time is not used as the primary determinant of product identity due to the precision of the MRM mode in isolating each product signal by mass and fragmentation pattern (Fig 3, S2 Table).

To calculate activity, a standard curve is first generated by plotting the area under the curve of the MRM signals against known concentration for each standard (Fig 3A). It is imperative that the standards do not only contain the desired deoxyribonucleoside(s) but also contain the same concentrations of substrates, effectors, and phosphates as the samples in order for standard curves to accurately reflect product concentration; the other components can affect ionization and thus detection of the deoxyribonucleoside standards in the mass spectrometer (S3 Fig). The areas under the curve of the MRM signals for the assay samples are then converted

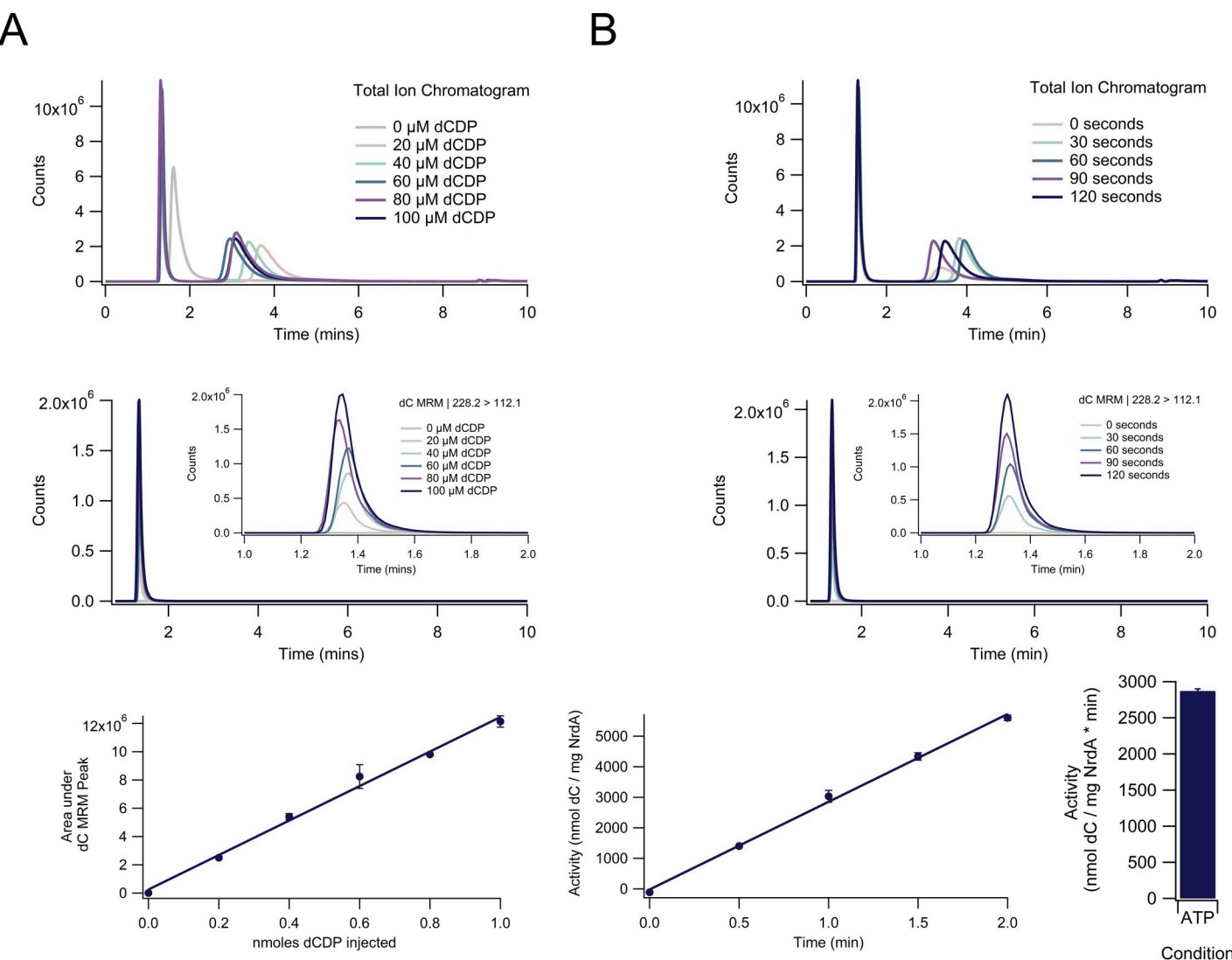

**Fig 3. Measurement of RNR activity for a single nucleotide under a single condition. (A)** Generation of a dC standard curve. A multiple reaction monitoring (MRM) curve for dC (m/z 228.2 > 112.1) is extracted from the initial total ion chromatogram (TIC) and the peaks are integrated (all in triplicate; one set of standards is shown above for the TIC and MRM) to generate a standard curve that relates the area under the dC MRM peak to the nmoles dC injected. The peak in the TIC at approximately 1.4 mins corresponds to the desired dephosphorylated product (dC) whereas the peak at approximately 3.5 mins corresponds to the dephosphorylated effector (A) in this experiment; note the MRM peak for dC is at the same retention time as the dC peak in the TIC. **(B)** Calculation of sample activity. Another set of MRM curves are extracted and integrated from TICs for each of the time points of the assay. The integrated MRM values are then converted to nmol dC / mg NrdA by using the standard curve to calculate nmol dC injected and then dividing by the mg NrdA in each injection. The slope of the line (in nmol/mg-min; after the aforementioned has been completed in triplicate) is the activity of the protein. All reactions are done in triplicate and are shown as mean ± standard error of the mean. The above condition contained NrdA (0.1 μM dimer), NrdB (0.5 μM dimer), ATP (3000 μM), CDP (1000 μM), Trx (30 μM), TrxR (0.5 μM), and NADPH (200 μM).

into quantity of product generated at each time point by using the standard curve (Fig 3B). The calculated product generated at each time point is graphed (in triplicate), and the slope of the resulting line (in nmol product/min) is used in conjunction with the mass of the limiting subunit of RNR (if applicable) to determine the reported activity in nmol product/mg-min.

## *E. coli* class Ia RNR activity assay results are similar to established activities

*Ec*RNR activities were measured using the LC-MS/MS assay under conditions found in the literature using the established radioactive and coupled assays. The activity of NrdA in the

presence of 5x NrdB, 1 mM substrate CDP, and 3 mM allosteric activator ATP was found to be 2880 ± 30 nmol dCDP/mg-min, whereas the activity of NrdB in the presence of 5x NrdA under the same nucleotide conditions was found to be 10000 ± 1000 nmol dCDP/mg-min (Fig 4). These data are in comparison to published values of 1500–3800 and 5000–8000 nmol dCDP/mg-min, respectively, using established (radioactivity and coupled) assays under the same nucleotide conditions [35, 51, 53, 54, 66–68]. In contrast, the activity of NrdA in the

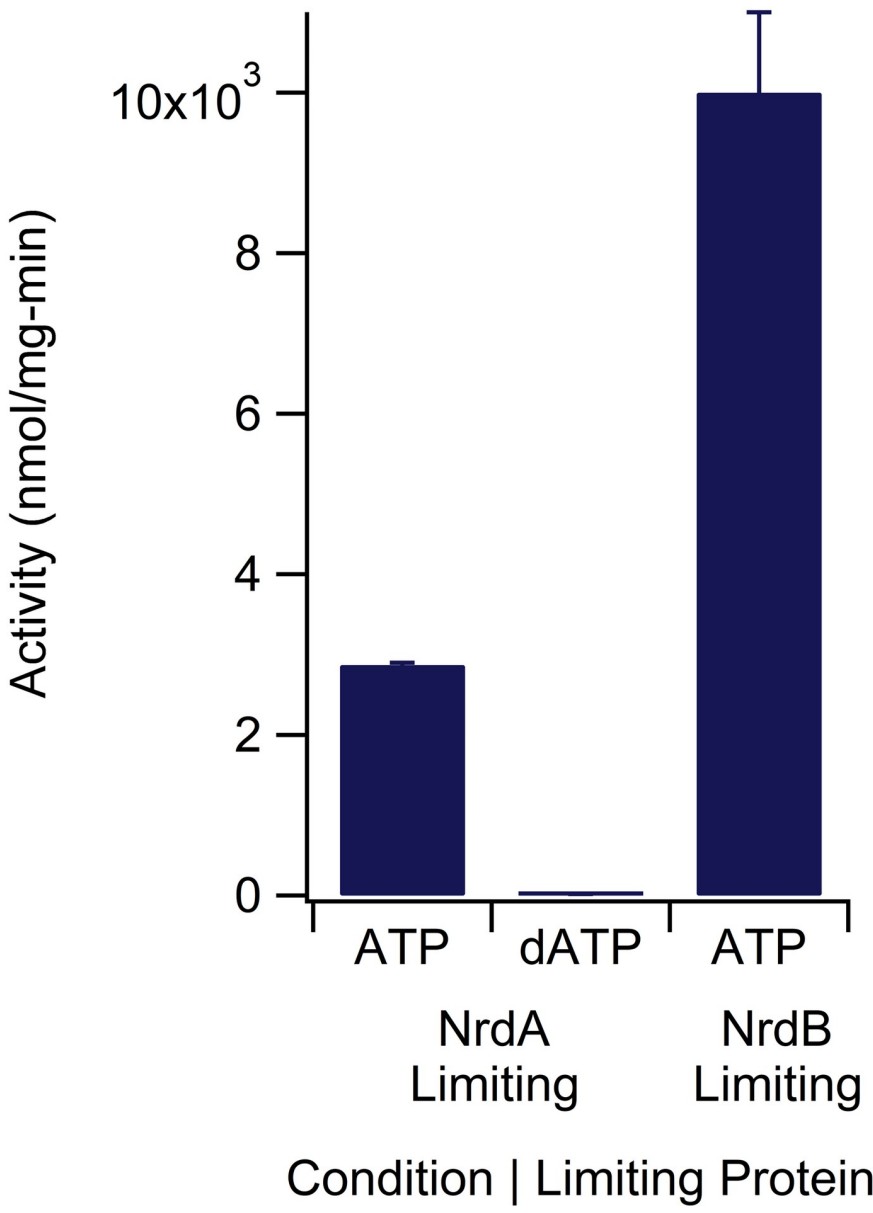

**Fig 4. Activity of *E. coli* class Ia ribonucleotide reductase.** Each bar represents average activity +/- standard error of the mean for three replicates. NrdA ATP and dATP conditions had 0.1 μM NrdA and 0.5 μM NrdB and 3000 μM ATP or 0.175 μM dATP, respectively, whereas the NrdB condition had 3000 μM ATP, 0.1 μM NrdB, and 0.5 μM NrdA. All conditions contained CDP (1000 μM), Trx (30 μM), TrxR (0.5 μM), and NADPH (200 μM). All protein concentrations are expressed as concentration of the dimer.

presence of 5x NrdB, 1 mM substrate CDP, and 0.175 µM allosteric inhibitor dATP was found to be 19 ± 1 nmol dCDP/mg-min, indicating 99% inactivation of NrdA by dATP (Fig 4). This activity is in comparison to published values of 175–300 nmol dCDP/mg-min using established assays under the same nucleotide conditions [35, 53]; similar studies on dATP inhibition of wild-type *E. coli* NrdA indicate that the addition of a similar concentration (100 µM) of dATP inhibits between 80% and >95% of enzyme activity as compared to activity in the presence of activating (~1 µM) concentrations of dATP and 0.5 mM CDP substrate [58, 69].

## Anaerobic (class III) RNR activity can be measured using the LC-MS/MS assay

The activity of the class III RNR from *S. thermophilus*, *St*NrdD, in the presence of 1 mM substrate GTP, 1 mM allosteric specificity effector TTP, and no allosteric activity effector was observed to be 110 ± 15 nmol dGTP/mg-min. In the presence of 3 mM allosteric activator ATP, 1 mM each GTP and TTP, activity was raised to 950 ± 70 nmol dGTP/mg-min. Activity was lowered to 3.9 ± 0.5 nmol dGTP/mg-min in the condition of 3 mM allosteric inhibitor dATP, 1 mM GTP, and 1 mM TTP (Fig 5). There are no previous determinations of *St*NrdD activity, but the activity of the class III RNR from *Lactococcus lactis* has been measured to be 800–1000 nmol dCTP produced/mg-min in the presence of 1.5 mM ATP and 0.2 mM dCTP; addition of dATP reduced activity to approximately 50 nmol dCTP/mg-min [39]. The T4 bacteriophage class III RNR, which does not exhibit activity regulation, has reported activities of 575 nmol dCTP produced/mg-min in the presence of 1.5 mM CTP and 1 mM ATP, 730 nmol/mg-min in the presence of 1.5 mM CTP and 1 mM dATP, and 370 nmol/mg-min in the presence of 1.5 mM CTP and no effector [36]. Other, much lower, activities have been reported in the literature for additional class III RNRs (10 ± 1, 49, and 55 ± 3 nmol dCTP produced/mg-min for class III RNRs from *Thermotoga maritima*, *Neisseria bacilliformis*, and *Methanosarcina barkeri*, respectively); however, these activities may increase with further optimization of fully anaerobic conditions, glycyl radical installation, and/or necessary assay components [28, 48, 70].

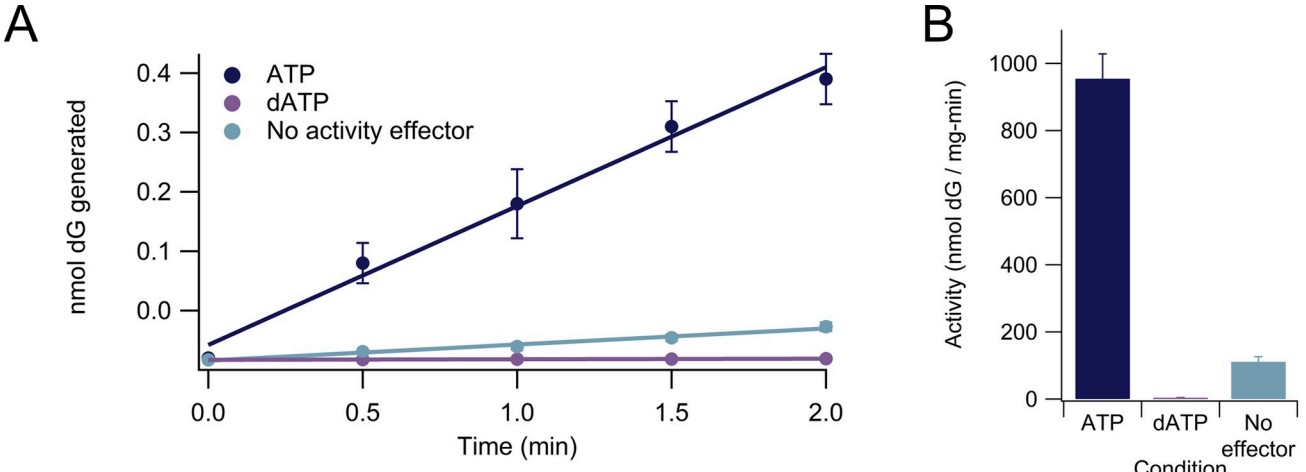

**Fig 5. Activity of *St*NrdD class III ribonucleotide reductase enzyme.** **(A)** Activity assay data for ATP, dATP, and no allosteric activity effector in the presence of GTP (substrate) and TTP (specificity effector). Error bars are shown for all replicates (3 replicates per data point) but are too small to visualize for some conditions. **(B)** Activities calculated from assay data in (A). Each bar represents average activity +/- standard error of the mean for three replicates. All conditions contained 0.10 µM *St*NrdD with 1.5 ± 0.10 glycyl radicals/dimer, 1 mM substrate GTP, 1 mM specificity effector TTP, 5 mM DTT, and 12.5 mM sodium formate; the ATP condition contained 3 mM allosteric activator ATP whereas the dATP condition contained 3 mM allosteric inhibitor dATP.

## An internal standard eliminates the need for same-day standard curves

All standard curves associated with data presented above were run on the same day as their associated samples due to day-to-day variability in the mass spectrometer. However, the need for same-day standard curves can be avoided with the addition of internal standards into both the standard curves and the assay samples (Fig 6). Addition of 1 μL of 1.6 mM heavy ($^{13}$C/$^{15}$N) dCTP to each sample and standard tube in a CDP reduction assay (leading to 0.4 nmoles heavy dC injected per assay), after the 95°C inactivation but prior to the addition of CIP, yields an internal standard peak for each sample and standard that can be used to eliminate day-to-day variability in the dC MRM peak integration value without significantly increasing the cost of the assay. The addition of a single heavy nucleotide as an internal standard adds approximately 25–50 cents per triplicate 5-timepoint assay. Despite two very different standard curves from the same dC standards that were generated over two different days due to instrument variability (Fig 6A), division of each day's dC MRM integration by the integration of the heavy

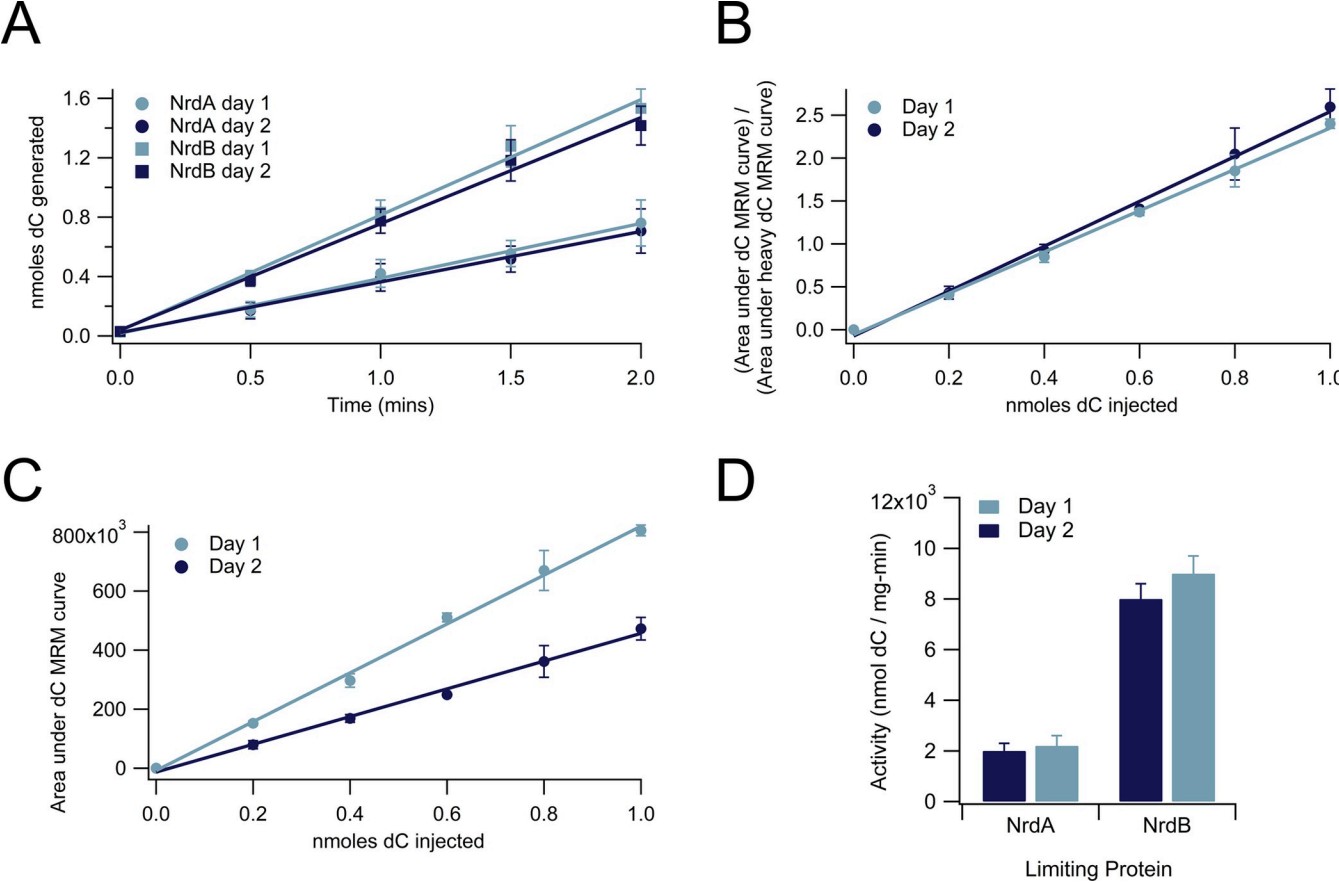

**Fig 6. The use of internal standards eliminates the need for same-day standard curves.** **(A)** Standard curves of the same amount of dC (0–1 nmole) run on the mass spectrometer on two different days (day 1 and day 2). **(B)** Standard curves run on day 1 and day 2 after internal standard correction (the area under the MRM curve of the heavy [$^{13}$C/$^{15}$N] dC nucleoside). **(C)** Activity assay data for *Ec*NrdA and *Ec*NrdB with nmoles dC calculated after correction of both standard and sample dC integration using a heavy dC internal standard. Both assays were run on day 2 along with the day 2 standard curve used for day 2 activity calculation; day 1 standard curve used for day 1 activity calculation was run on a previous day. **(D)** Calculation of activity from curves in (C) with *Ec*NrdA or *Ec*NrdB as the limiting reagent. NrdA activity was determined to be 2000 ± 300 nmoles dC/mg-min using the day 2 standard curve to calculate activity and 2200 ± 400 nmoles dC/mg-min using the day 1 standard curve to calculate activity. NrdB activity was determined to be 8000 ± 600 nmoles dC/mg-min using the day 2 standard curve to calculate activity and 9000 ± 700 nmoles dC/mg-min using the day 1 standard curve to calculate activity. The above conditions contained NrdA (0.1 μM dimer for NrdA condition and 0.5 μM dimer for NrdB condition), NrdB (0.5 μM dimer for NrdA condition and 0.1 μM dimer for NrdB condition), ATP (3000 μM), CDP (1000 μM), Trx (30 μM), TrxR (0.5 μM), and NADPH (200 μM).

internal standard led to two almost identical standard curves (Fig 6B). When used to calculate nmoles dC generated in two different activity assays (one with *Ec*NrdA as the limiting protein and one with *Ec*NrdB as the limiting protein), the corrected standard curves yielded almost identical activity curves (Fig 6C). These activity curves led to indistinguishable calculations of final activity for *Ec*NrdA (2000 ± 300 vs. 2200 ± 400 nmoles dC/mg-min) and *Ec*NrdB (8000 ± 600 nmoles dC/mg-min vs. 9000 ± 700 nmoles dC/mg-min).

## All four ribonucleotide substrates can be assayed simultaneously using the LC-MS/MS assay

In addition to assaying the production of a single deoxyribonucleotide product, the production of all four deoxyribonucleotide products (dADP, dCDP, dGDP, and dUDP for class I RNRs) can be simultaneously measured using the LC-MS/MS assay (Fig 7; S4 Fig). The total ion chromatogram (TIC) containing all four deoxyribonucleoside traces can be deconvoluted using multiple reaction monitoring (MRM) for each of the four deoxyribonucleoside products to generate four separate activity assay curves (Fig 7; S4 Fig).

We determined the activity of *E. coli* NrdA on all four ribonucleotide substrates in the presence of approximately physiological concentrations of substrate and effector nucleotides with varying concentrations (0–1 mM) of each specificity effector (Fig 8, Table 2) [71, 72]. Our estimates of physiological nucleotide concentrations were based on measurements made by Buckstein and colleagues in mid-log growth for *E. coli*: ATP (3560 μM), dATP (181 μM), dGTP (54 μM), TTP (256 μM), ADP (116 μM), GDP (203 μM), UDP (54 μM) [72]; these concentrations also closely match previously-published values [73]. We also considered Traut et al. values for non-human organisms, especially for CDP concentrations which were not measured in the studies listed above: ATP (3000 ± 2000 μM), dATP (20 ± 20 μM); dGTP (5 ± 4 μM), TTP (40 ± 30 μM), ADP (800 ± 400 μM), CDP (70 ± 50 μM), GDP (160 ± 50 μM), UDP (200 ± 100 μM) [71].

We first investigated dATP (physiological concentrations in mid-log phase *E. coli* reported of 24, 175, and 181 μM [71–73]) as the variable effector. dATP is known to be an allosteric specificity effector for CDP and UDP at lower concentrations and an allosteric inhibitor at higher concentrations (see Fig 1C). We found that increasing the amount of dATP from 0 to 200 μM, raising it closer to physiological levels, in the presence of 3000 μM ATP, 100 μM dGTP, and 250 μM TTP raised the overall activity of the enzyme on the ribonucleotide pool

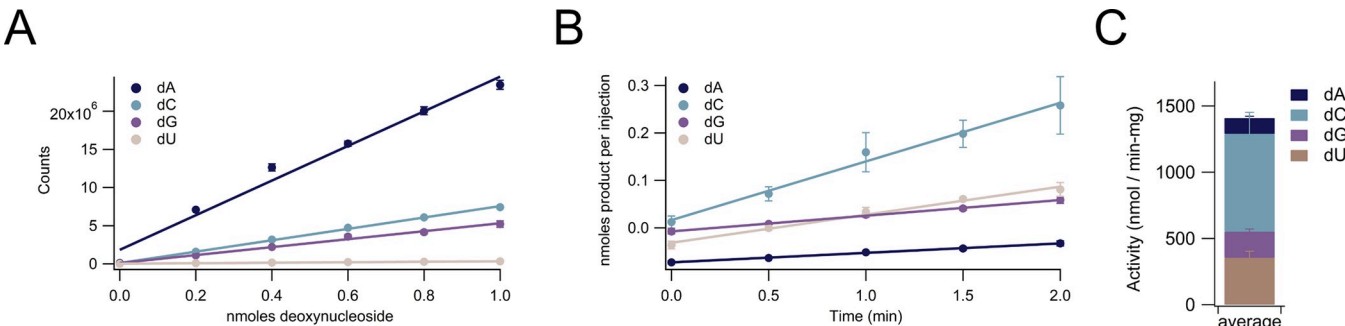

**Fig 7. Activity of *E. coli* class Ia ribonucleotide reductase reducing all four ribonucleotides simultaneously in the presence of specificity effectors dATP, dGTP, and TTP and activity effector ATP. (A)** Standard curves of all four ribonucleotides mixed together in concentrations of 0–100 μM (10 μL injections). **(B)** Activity graphs of all four ribonucleotide products being produced simultaneously over two minutes. **(C)** Bar graph showing production of each of the four deoxyribonucleotide products. The dU activity above condition contained Trx (30 μM), TrxR (0.5 μM), NADPH (200 μM), ATP (3000 μM), dATP (500 μM), TTP (250 μM), dGTP (100 μM), CDP (70 μM), UDP (50 μM), GDP (200 μM), ADP (110 μM), NrdA (0.1 μM dimer), and NrdB (0.5 μM dimer). Total ion chromatogram (TIC) and individual multiple reaction monitoring (MRM) curves are shown in S4 Fig.

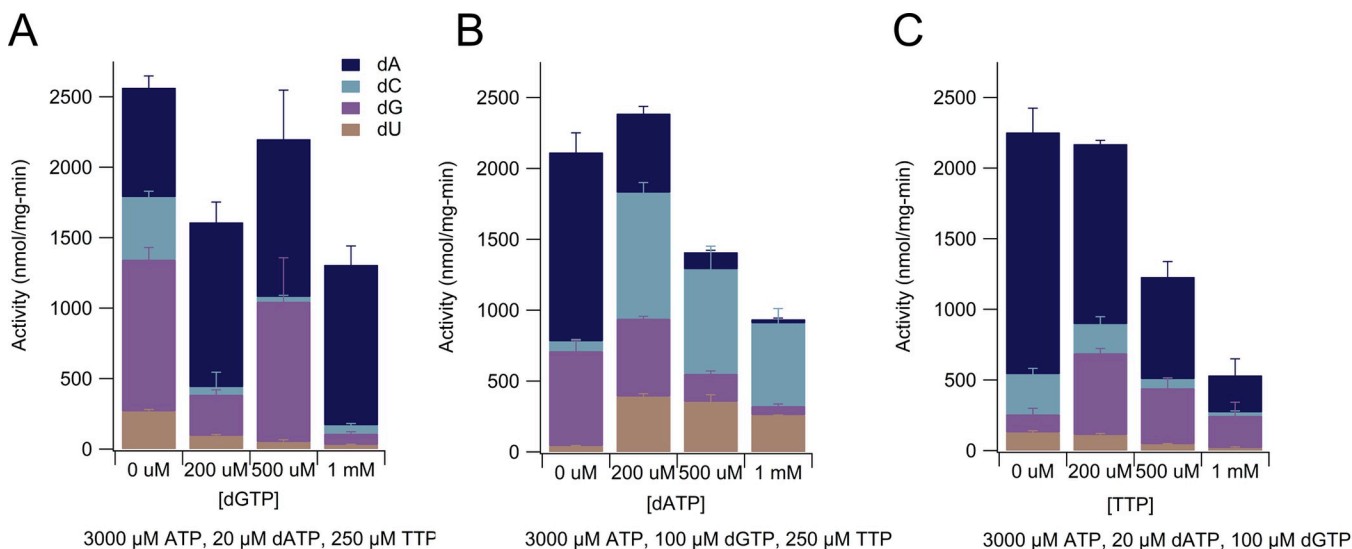

**Fig 8. Assay data with class Ia *E. coli* ribonucleotide reductase measuring the reduction of all four ribonucleotide substrates simultaneously. (A)** Activity of *Ec*NrdA in the presence of substrates, effectors, and varying amounts of dATP. **(B)** Activity of *Ec*NrdA in the presence of substrates, effectors, and varying amounts of dGTP. **(C)** Activity of *Ec*NrdA in the presence of substrates, effectors, and varying amounts of TTP. Each bar is representative of the average activity +/- standard error of the mean for three replicates. The values of each bar are shown in Table 2. Except for the indicated concentrations of allosteric effectors on the X-axis, components and concentrations were as follows: Trx (30 μM), TrxR (0.5 μM), NADPH (200 μM), CDP (70 μM), UDP (50 μM), GDP (200 μM), ADP (110 μM), NrdA (0.1 μM dimer), NrdB (0.5 μM dimer).

(CDP [70 μM], UDP [50 μM], GDP [200 μM], ADP [110 μM]) from 2100 ± 100 to 2380 ± 90 nmol nucleotides reduced/mg-min (Fig 8A, Table 2). This overall modest activity increase is due to large increases in dCDP and dUDP production offset by a large decrease in dADP production and a modest decrease in dGDP production (Fig 8, Table 2). Notably, at 0 mM dATP and approximately physiological concentrations of other effectors, dCDP and dUDP production comprised only 5% of the total enzyme activity, whereas that percentage increased to 54% at 200 μM dATP and increased further to 75% at 500 μM dATP and 86% at 1 mM dATP (Fig 8A, Table 2). Despite the percentage increase for dCDP and dUDP compared to dADP and dGDP with the increase in dATP concentrations from 200 μM to 500 μM and then to 1 mM, enzyme activity on all four ribonucleotide substrates, including CDP and UDP, decreases with increasing concentrations of dATP past average reported physiological levels (Fig 8A, Table 2). Instead of a total enzyme activity of 2380 ± 90 nmol nucleotides reduced/mg-min at 200 μM dATP, at 1 mM dATP, the overall activity is 1400 ± 300 nmol nucleotides reduced/mg-min (Fig 8A, Table 2).

Using the LC-MS/MS assay, we were also able to investigate the degree to which higher concentrations of dATP are needed to inhibit *E. coli* NrdA's activity on CDP when other effectors are in the mixture. In particular, NrdA's activity on CDP with dATP alone (175 μM dATP and 1 mM CDP) is almost nonexistent (Fig 4) compared to an activity value of 890 ± 70 nmol nucleotides reduced/mg-min with 200 μM dATP in the presence of approximately physiological concentrations of ATP (3000 μM) and substrates and other specificity and activity effectors (70 μM CDP, 50 μM UDP, 200 μM GDP, 110 μM ADP, 100 μM dGTP, and 250 μM TTP) (Fig 8A, Table 2). Although we expected ATP to counter the inhibitory effect of dATP, using this assay we can quantitate the effect of ATP/dATP ratios on RNR activity.

When dGTP (physiological concentrations in mid-log phase *E. coli* reported of 5, 92, and 122 μM [71–73]) is the variable specificity effector, an increase from 0 to 200 μM in dGTP in the presence of 3000 μM ATP, 20 μM dATP and 250 μM TTP results in an increase in dADP

**Table 2. Average activities +/- standard error of the mean of assays graphically represented in Fig 8.** In addition to the indicated concentration of the allosteric effectors shown below in column 1, components and concentrations were as follows: Trx (30 μM), TrxR (0.5 μM), NADPH (200 μM), CDP (70 μM), UDP (50 μM), GDP (200 μM), ADP (110 μM), NrdA (0.1 μM dimer), NrdB (0.5 μM dimer).

| Allosteric effectors | Concentration of variable effector | dCDP production (nmol/mg-min) | dUDP production (nmol/mg-min) | dADP production (nmol/mg-min) | dGDP production (nmol/mg-min) | Overall activity (nmol/mg-min) |
|---|---|---|---|---|---|---|
| Variable dATP | 0 mM | 70 ± 10 | 41 ± 4 | 1300 ± 100 | 670 ± 80 | 2100 ± 100 |
| 3000 μM ATP | 200 μM | 890 ± 70 | 390 ± 20 | 550 ± 50 | 550 ± 20 | 2380 ± 90 |
| 100 μM dGTP | 500 μM | 700 ± 200 | 350 ± 50 | 120 ± 10 | 200 ± 20 | 1400 ± 300 |
| 250 μM TTP | 1000 μM | 600 ± 100 | 260 ± 1 | 28 ± 9 | 60 ± 20 | 1000 ± 100 |
| Variable dGTP | 0 mM | 440 ± 40 | 270 ± 10 | 780 ± 80 | 1080 ± 90 | 2600 ± 100 |
| 3000 μM ATP | 200 μM | 50 ± 100 | 94 ± 9 | 1200 ± 100 | 290 ± 30 | 1600 ± 100 |
| 20 μM dATP | 500 μM | 30 ± 10 | 50 ± 20 | 1100 ± 300 | 1000 ± 300 | 2200 ± 400 |
| 250 μM TTP | 1000 μM | 60 ± 10 | 31 ± 4 | 1100 ± 100 | 80 ±10 | 1300 ± 100 |
| Variable TTP | 0 mM | 280 ± 40 | 130 ± 10 | 1700 ± 200 | 130 ± 40 | 2200 ± 200 |
| 3000 μM ATP | 200 μM | 210 ± 50 | 110 ± 10 | 1280 ± 30 | 580 ± 30 | 2180 ± 70 |
| 20 μM dATP | 500 μM | 64 ± 9 | 45 ± 4 | 700 ± 100 | 400 ± 70 | 1200 ± 100 |
| 100 μM dGTP | 1000 μM | 20 ± 10 | 19 ± 8 | 300 ± 100 | 200 ± 100 | 600 ± 100 |

production and a decrease in dCDP, dUDP, and dGDP production, as expected (see Fig 1C). Enzyme activity on substrate ADP levels off above 200 μM dGTP as levels of dGTP increase further from physiological levels, but the percentage of dADP made compared to dCDP, dUDP, and dGDP is highest at the highest concentration of dGTP (1 mM) (Fig 8, Table 2). When TTP (physiological concentrations in mid-log phase *E. coli* reported of 37, 256, and 77 μM [71–73]) is the variable specificity effector, an increase from 0 to 200 μM in TTP in the presence of 3000 μM ATP, 20 μM dATP and 100 μM dGTP results in an increase in dGDP production and modest decreases in dCDP, dUDP, and dADP production (Fig 8, Table 2). Again, these results are consistent with the specificity rules as shown in Fig 1C. Enzyme activity on substrate GDP reaches a maximum at the approximately physiological concentration of 200 μM TTP (580 +/- 30 nmol/mg-min) and then begins to decrease (Fig 8, Table 2).

In general, increasing amounts of any of the effectors to a supraphysiological (1 mM) concentration dampened overall activity of the enzyme, with the enzyme displaying 48%, 50%, and 27% of its initial (0 mM effector) activity when dATP, dGTP, or TTP, respectively, were raised to a 1 mM concentration (Fig 8, Table 2).

## Discussion

The ribonucleotide reductase field has been challenged by a lack of accessible and validated assays that are flexible enough to be used to measure activity of all classes of RNRs. The assay

**Table 3. Comparison of the radioactivity and coupled RNR activity assays and the LC-MS/MS assay described in this paper.**

|  | *Radioactivity Assay* | *Coupled Assay* | *LC-MS/MS Assay* |
|---|---|---|---|
| **Safety Considerations** | Radioactivity license needed | None | None |
| **Substrates** | CDP or CTP primarily | Any | Any |
| **Reducing Systems Accommodated** | Any | NADPH only | Any |
| **Classes of RNRs Assayable** | Any | Class I and II, theoretically some class III | Any |
| **Multiple Substrates Assayable Simultaneously?** | No | No | Yes |
| **Equipment Required** | Scintillation Counter | UV-Vis Spectrophotometer | QQQ Mass Spectrometer |

presented here does not have the safety considerations of the radioactive assay and is more flexible in the substrates and reducing systems, and thus increases the classes of RNRs that can be assayed relative to the radioactivity or coupled assays (Table 3). It is also versatile in that various ratios of NrdA:NrdB (for class I RNRs), different substrates/effectors or combinations of substrates/effectors, and different temperatures can be assayed. Additionally, the assay is higher throughput than some existing assays in that many LC-MS vials can be prepared over the course of days and then run with short run times using an autosampler, allowing for activity under many conditions to be determined in a short amount of time. The assay presented here improves on existing HPLC-based assays through the use of short run times, a lack of need to fully deconvolute or identify overlapping or contaminant peaks in the HPLC due to the use of mass spectrometry MRM for specific product parent and fragment ion peaks, and by being thoroughly validated using known proteins and their associated activities prior to moving on to examination of novel RNRs.

*E. coli* class Ia RNR (NrdA/NrdB) is the most well studied RNR and has been characterized since the early 1960s [9, 10, 42]. Thus, for the purposes of validation, there is a large body of work determining protein activities that can be compared to the activities found using the assay presented here. Although activities can vary slightly due to radical content, protein purity, the method of determining protein concentration, and age of the protein being assayed among other properties and protocols, the activities of both *E. coli* NrdA and NrdB as determined by the current assay under ATP and dATP conditions closely match those reported in the literature [35, 51, 53, 54, 66–68]. The agreement between the activity numbers measured with this assay and the past literature numbers serves to support the validity of this assay. We note that the *E. coli* proteins studied here are all N-terminally His-tagged, which for NrdA and NrdB has been previously shown to not interfere with activity [74]; however, for characterization of new RNRs we recommend determining activity with and without a tag to ensure the tag is not interfering in protein activity. The Trx and TrxR used here are also His-tagged. To mitigate any decrease in activity, we used a 60x fold excess of Trx in our assays to ensure re-reduction was not the limiting step in any activity assay.

In comparison to the commonly studied *E. coli* class Ia RNR, there are numerous RNRs that are un- or under-characterized due to the conditions under which they function. This understudied group includes the class III RNRs, which must be studied under anaerobic conditions due to their use of a highly oxygen-sensitive glycyl radical. Additionally, many class III RNRs use formate, not NADPH, as a terminal electron acceptor, making the coupled assay impossible [75]. The radioactivity assay also proves difficult for class III RNRs due to the requirements for an anaerobic chamber that can accommodate radioactive samples and the inherent challenges of performing an assay in an anaerobic environment. Furthermore, class III RNRs have NTPs as both substrates and effectors, as opposed to the NDP substrates and NTP effectors of the well-studied class I enzymes. This dual use of NTPs can create challenges in the separation of effectors and products in some assays. The LC-MS/MS-based assay described here overcomes the above challenges and allows for the determination of activity of any class III RNRs as long as the lab is equipped with an anaerobic chamber. Here we use this LC-MS/MS-based assay to show that the *St*NrdD, a class III RNR that is expected by sequence to have allosteric activity regulatory or cone domains, is in fact allosterically regulated by ATP and dATP and displays activity on an order of magnitude similar to what has been shown with other class III RNRs [36, 39]. ATP enhances activity and dATP inhibits this class III enzyme. To the best of our knowledge, this is only the third time that a class III RNR has been shown experimentally to be allosterically regulated by ATP and dATP. Previously, the class III RNRs from *E. coli* and *Lactococcus lactis*, both of which contain cone domains, have been observed to be under allosteric activity enhancement by ATP and inhibition by dATP [39, 76].

A distinct advantage of the presented assay over the assays currently available to the RNR field is that the LC-MS/MS assay is able to measure the formation of all four deoxyribonucleotide products simultaneously. Although there is precedent in the literature for HPLC assays that can measure all four products simultaneously, those assays with published total run times and flow rates take 45–180 minutes per run at a flow rate of 0.4–1.5 mL/min, placing buffer and time limitations on the number of samples that can be run as compared to the assay presented here (which takes 10 minutes per run at a flow rate of 0.350 mL/min) [44, 60].

Using the ability to measure all four dNDP products at once, we were able to show how increasing the concentration of one specificity effector at a time while keeping the other specificity effectors concentration constant affected the ratios of the dNDP products under a given set of conditions. When the concentration of any one specificity effector was increased from zero to approximately physiological levels, we saw an increase in enzyme activity on the substrates known to be the cognate pair(s) of that specificity effector; RNR activity on CDP and UDP increased when 200 μM dATP was added and the ratios of dCDP/dUDP to dADP and dGTP increased; RNR activity on ADP increased with addition of 200 μM dGTP and the ratio of dADP to other dNDPs increased considerably; and RNR activity on GDP increased with addition of 200 μM TTP although the combination of 200 μM TTP, 3 mM ATP, 0.2 mM dATP, and 0.10 mM dGTP makes for an RNR that is quite active on ADP substrate as well (Table 2; Fig 8). As previously reported, the presence of specificity regulators increases RNR activity in the 1.5–10 fold range [76, 77]. These data help us appreciate how an increase in the concentration of one effector can impact the ratios of all deoxyribonucleotides in a cell and the importance of cellular processes that keep all nucleotides within the physiological range.

We also see that large increases past the normal physiological range of any effector negatively impact RNR activity. Although dATP is a known allosteric activity inhibitor [78, 79], high concentrations (1 mM) of both dGTP and TTP also dampened overall EcRNR activity on its substrates. The dampening affected the reduction of all four ribonucleotide substrates under variable dATP and TTP conditions, whereas the decrease was not seen for the ADP substrate under variable dGTP conditions but was seen for the other three substrates. The mechanism of activity reduction is unlikely to be due to dGTP and TTP binding to the allosteric activity site, as that site is specific for dATP and ATP [41, 42]. The finding that an imbalance in the deoxyribonucleotide pool can affect overall RNR enzyme activity, which consequently would further impact the overall size and distribution of deoxyribonucleotides in the cell, shows how easily one imbalance can lead to further potentially disastrous imbalances in the cellular environment [29].

We additionally saw other impacts of measuring enzyme activity under pseudo-physiological nucleotide conditions instead of only supplying the enzyme with the necessary components to complete the desired reaction. We found that the concentration of dATP needed to inhibit E. coli class Ia RNR is much higher when ATP is also present at approximately physiological conditions. These results underscore the importance of having the protein in an approximately physiological nucleotide milieu when making conclusions to which one wishes to extend biological relevance.

Many examples of the assay presented here use a standard curve that must be run in triplicate before each set of assays due to day-to-day variability in the mass spectrometer; however, in the case of many assays being done over days or weeks over the course of an experiment, heavy deoxyribonucleotide dNTP standards (0.4 nmoles / injection) can be added to each standard and reaction mixture after the 95°C inactivation step but prior to dephosphorylation via CIP. The internal standards used must match the nucleotide identity of the product being measured in order to closely match the fragmentation and flight path of the desired products. Because the four deoxyribonucleotide assays already use heavy nucleotides as effectors (since

dNTPs and dNDPs cannot be distinguished after dephosphorylation), internal standards via heavy nucleotides are not an option for the four deoxyribonucleotide assays, but are options for any single ribonucleotide assay; these assays are still feasible by running same-day standard curves on the mass spectrometer, which can be formulated in advance in volumes that facilitate running many assays and stored stably at room temperature or -20˚C prior to use.

The assay presented is a flexible one, allowing for differences in protein concentration, temperature, and reducing agents to be easily accommodated. This flexibility means that a wide variety of RNRs, including the *E. coli* and *S. thermophilus* RNRs presented here, can be assayed using this protocol or a modification of this protocol. Proteins such as the *S. thermophilus* RNR that are thermophilic should be assayed for temperature of unfolding before using heat as a denaturant; however, even enzymes that require very high temperatures for inactivation and may not inactivate immediately upon insertion into the 95˚C heat block can still be assayed. The delay on inactivation will be constant across time points, and so the slope of the product generation line that determines enzyme activity will be the same regardless. Alternatively, heating-independent methods for stopping the reaction can be employed such as acid, methanol, or cold [11, 21, 44, 58, 59, 61, 63]. To verify that the method used to stop the reaction is satisfactory and that no refolding of the enzyme has occurred, the enzyme can be assayed following "inactivation" to ensure that there is no measured activity.

Our lab has successfully assayed other RNRs, including the *Neisseria gonorrhoeae* and human class Ia RNRs, using this assay; these data will be presented in future publications. Additionally, in the absence of physiological reducing agents or during experiments where cloning and purification of one or many organismal Trx and TrxR proteins is not feasible, 10–50 mM dithiothreitol (DTT) can be used as a universal reductant for class Ia RNRs. However, the efficiency of DTT can vary appreciably and optimization of DTT concentration should be performed for each enzyme and enzyme variant [80–82]. Finally, the linear range of each of the deoxyribonucleosides is at least in the femtomole or picomole to nanomole range (S5 Fig), meaning that a wide range of activities can be measured.

Future applications of LC-MS/MS for the study of RNRs include determinations of the stoichiometries of nucleotides binding to RNRs and their associated Kds. This assay could be adapted to be less labor intensive through the substitution of manual steps, following generation of the mastermix, with automated liquid handling. We look forward to seeing the ways that this assay is employed to investigate both well-characterized and under-characterized RNRs.

## Materials and methods

Reagents were procured from MilliporeSigma unless otherwise noted.

### Protein expression and purification

***E. coli* proteins (NrdA, NrdB, Trx, TrxR).** *E. coli* NrdA, NrdB, Trx, and TrxR were expressed from codon-optimized pET30(a) vectors as N-terminally His-tagged proteins (EMD; Quintara Biosciences). Expression of all proteins began with a 1:100 inoculation of overnight culture grown from a single colony of T7 Express cells (NEB) into Luria Broth (LB) + 50 ng/μL kanamycin (GoldBio). The cultures were grown at 37˚C with shaking at 220 rpm until $OD_{600}$ reached 0.6–0.8, at which point the cells were moved to 18˚C before induction with 1 mM isopropyl ß-D-1-thiogalactopyranoside (IPTG) (Gold Bio). Cells continued shaking overnight (approx. 20–22 hours) before being spun down at 4000 x g for 15 minutes at 4˚C and frozen at -20˚C until lysis and purification. To begin purification, cells were thawed on ice and resuspended in 30 mL lysis buffer (50 mM sodium phosphate monobasic pH 8.0, 500 mM

sodium chloride, 10 mM imidazole, 10% glycerol (VWR), 5 mM beta-mercaptoethanol for NrdA and NrdB; 50 mM tris(hydroxymethyl)aminomethane (Tris) pH 7.6, 300 mM sodium chloride, 30 mM imidazole, 5 mM beta-mercaptoethanol for Trx and TrxR) per 2 L cells. The following was added to the resuspension: 0.1% Triton X-100, one ethylenediaminetetraacetic acid (EDTA)-free protease inhibitor tablet (Roche), 5 μL benzonase, and a small amount (~5 mg) of lysozyme. Once cells were completely resuspended, they were sonicated on ice for 3 minutes and 30 seconds at 70% amplitude with 1 second of sonication on followed by 3 seconds of sonication off. Lysed cells were spun down at 25000 x g for 30 minutes at 4°C before the clarified lysate was loaded onto a 5 mL HisTrap column (GE Life Sciences) equilibrated with lysis buffer. The lysate was batch-bound for one hour using a peristaltic pump running at 1 mL/min before flow-through was collected. The protein was then eluted off of the HisTrap column using a gradient from 4–100% B (B = lysis buffer + 500 mM imidazole) at 2.5 mL/min. Fractions containing the protein of interest were determined using SDS-PAGE (Bio-Rad, 4–20% gel, 200 V) and then were concentrated using a 30 KDa (NrdA and NrdB) or 10 kDa (Trx and TrxR) spin concentrator (Millipore) to a final volume of 1–3 mL. Concentrated protein was injected onto a Superdex S200 16/60 column (GE Life Sciences) equilibrated with 50 mM 4-(2-hydroxyethyl)-1-piperazineethanesulfonic acid (HEPES) pH 8, 100 mM potassium chloride, 15 mM magnesium chloride, and 10% glycerol for NrdA and NrdB (with the addition of 1 mM DTT [Research Products International] for NrdA) or 20 mM HEPES pH 7.6, 100 mM sodium chloride, 5% glycerol for Trx and TrxR. Purity of the compounds was determined by running an SDS-PAGE gel (4–20%, 200 V; expected molecular weights 85774, 44528, 13612, and 36429 Da for NrdA, NrdB, Trx, and TrxR, respectively). Concentrations of the proteins were determined by $A_{280}$ ($\varepsilon$ = 88590, 62340, 15470, and 20400 $M^{-1}cm^{-1}$ for NrdA, NrdB, Trx, and TrxR, respectively [per monomer]). Concentration of flavinated TrxR was subsequently determined at 458 nm ($\varepsilon$ = 11200 $M^{-1}cm^{-1}$), which is the molar absorptivity of the flavin cofactor. If necessary, proteins were further concentrated using a 30 kDa (NrdA and NrdB) or 10 kDa (Trx and TrxR) spin concentrator (Millipore) prior to being aliquoted and flash-frozen in liquid nitrogen for storage at -80°C.

Reconstitution of the NrdB di-iron cluster was conducted in an anaerobic (100% nitrogen atmosphere; Airgas or Medtech) MBraun chamber held at 4°C as previously described [51, 78]. Aerobic RNR storage buffer (50 mM HEPES pH 8, 100 mM KCl, 15 mM magnesium chloride, 10% glycerol) was sparged for 30 minutes with argon gas (Airgas or Medtech) and brought into the anaerobic chamber along with approximately 10 mg ferrous ammonium sulfate powder and the protein to be reconstituted (with headspace degassed). Once inside the chamber, the iron ammonium sulfate was dissolved in 1 mL storage buffer and enough iron ammonium sulfate was added dropwise to the protein solution with stirring to reach 5 molar equivalents iron per beta dimer. The reaction was allowed to stir in the chamber for 15 minutes before it was removed from the anaerobic chamber and immediately mixed with oxygen-saturated storage buffer generated by bubbling oxygen (Airgas) through the buffer for 10 minutes prior to use. The volume of oxygen-saturated buffer added equaled 10 molar equivalents of oxygen per beta dimer using the assumption that the concentration of oxygen in saturated buffer is 1.4 mM. Oxygen was then blown over the surface of the protein solution for 2 minutes before protein was spun down at 13,000 x g for 1 minute to remove precipitate. The protein was then applied to a HiPrep 26/10 desalting column (GE Life Sciences) equilibrated with RNR storage buffer in order to remove excess iron from the protein solution before concentration and storage of the protein at -80°C. Radical content was determined to be 1.2 radicals per reconstituted NrdB dimer by electron paramagnetic resonance (EPR; Bruker EMX-Plus) (S6 Fig). EPR parameters were 80K, 10 scans, modulation amplitude 1.5G, center field 3330 G, sweep width 300G, sweep time 20 s, microwave power 38dB.

**S. thermophilus proteins (NrdD, NrdG).**    *S. thermophilus* NrdD, originally amplified from a commercial yogurt sample, was cloned into and expressed from a pET15(b) vector as N-terminally His$_6$-tagged protein with a thrombin cleavage sequence in T7Express cells (NEB). Expression began with a 1:200 inoculation of overnight culture grown from a single colony into LB with 50 μM ampicillin (Research Products International) and 50 μM zinc sulfate. The cultures were grown at 37°C with shaking at 220 rpm until OD$_{600}$ reached 0.6–0.8, at which point the cells were moved to 16°C before induction with 0.5 mM IPTG. Cells continued shaking overnight (approx. 18–20 hours) before being spun down at 4000 x g for 15 minutes at 4°C. The resulting cell pellet was frozen at -80°C until lysis and purification. To begin purification, cells were thawed at room temperature and resuspended in 30 mL buffer A (50 mM HEPES pH 7.6, 300 mM sodium chloride, 1 mM tris(2-carboxyethyl)phosphine [TCEP]) per 2 L cells. The following was added to the resuspension: 0.1% Triton X-100, 0.5 mM phenyl-methylsulfonyl fluoride (PMSF), one EDTA-free protease inhibitor tablet (Roche), 5 μL benzonase, and a small amount (~5 mg) of lysozyme. Once cells were completely resuspended, they were nutated at 4°C for 15 min before lysis by sonication on ice for 3 minutes and 30 seconds at 70% amplitude with 1 second of sonication on followed by 3 seconds of sonication off. Lysed cells were spun down at 25000 x g for 40 minutes at 4°C before the clarified lysate was loaded onto a 5 mL HisTrap column equilibrated with buffer A. The lysate was batch-bound for thirty minutes using a peristaltic pump running at 1 mL/min before flow-through was collected. The column was first washed, at a rate of 1.5 mL/min, with 10 column volumes of buffer A, followed by a 5-column volume wash with buffer A supplemented with 10 mM imidazole. The protein was eluted off the HisTrap column using buffer A supplemented 300 mM imidazole. Fractions containing the protein of interest were determined using SDS-PAGE (4–20% gel, 200 V), and were diluted two-fold with buffer A before concentrating using a 50 kDa spin concentrator (Millipore) to a final volume of 3–5 mL. Concentrated protein was injected onto an S200 16/60 column equilibrated with 20 mM HEPES pH 7.6, 100 mM sodium chloride. Purity of the compounds was determined by running an SDS-PAGE gel (4–20%, 200 V; expected molecular weight 85619 Da). Concentrations of the proteins were determined by A$_{280}$ (ε = 189010 M$^{-1}$cm$^{-1}$ [per dimer]). Proteins were further concentrated using a 50 kDa spin concentrator (Millipore) prior to being aliquoted and flash-frozen in liquid nitrogen for storage at -80°C.

*S. thermophilus* NrdG was expressed from a pET28(a) vector as a codon optimized His$_6$-tagged protein with a thrombin cleavage site (codon optimized sequence WP_011681737.1 synthesized by Genscript) in BL21 (DE3) cells (Novagen). Cultures, supplemented with kanamycin at 50 ng/μL (GoldBio), were grown to an OD600 of 0.6 before induction with 1 mM IPTG. Cells continued shaking overnight at 16°C. Cells were pelleted by centrifugation and resuspended in 10 mL/1L of culture in 50 mM HEPES pH 7.6, 300 mM sodium chloride, 5% glycerol, 1 mM TCEP, and 0.5 mM PMSF. Cells were lysed by sonication on ice for 4 minutes at 70% amplitude with 1 second of sonication on followed by 3 seconds of sonication off. The lysate was clarified by centrifugation at 25,000 x g for 40 minutes in the presence of 1 μL benzonase per 1 L of culture. Lysate was filtered and applied to 10 mL of TALON resin (GE Life Sciences), then washed with 20 mM HEPES pH 7.6, 100 mM potassium chloride, 1% (v/v) glycerol supplemented with 15 mM imidazole. Protein was eluted with the 20 mM HEPES pH 7.6, 100 mM potassium chloride, 1% (v/v) glycerol supplemented with 150 mM imidazole. Protein was buffer exchanged to remove imidazole and glycerol, then concentrated. His$_6$-tag cleavage was completed next by incubation at 4°C overnight with 1 U of thrombin (GE Life Sciences) per mg of protein. Thrombin was quenched by addition of 0.5 mM PMSF and tag-free protein was isolated by subtractive TALON. Tag-free protein was concentrated to ~10 mg/mL (using Bradford correction value of 0.69), sealed in a bottle, and the headspace

degassed by sparging with argon gas. Protein was brought into an anaerobic chamber with a 100% Nitrogen atmosphere (MBraun) maintained at 4˚C, opened, and DTT added to 5 mM. Protein sat vented for 5 days in the MBraun at 4˚C before the entire stock was reconstituted.

**$St$NrdG reconstitution.** Before reconstitution in the MBraun chamber, DTT was added to a final concentration of 10 mM and protein was diluted with 20 mM HEPES pH 7.6, 100 mM KCl, 10 mM DTT (Buffer A1) to a concentration of 100 uM monomer (~2 mg/mL). Reconstitution was initiated by adding 5 equivalents of iron (III) chloride ($FeCl_3$), then sodium sulfide ($Na_2S$), swirling after each addition and incubating for 4 hours. Reconstituted protein was spun quickly, concentrated, then desalted by anaerobic size exclusion on an 16/60 Superdex 200 (GE Lifesciences) using buffer A1. This procedure yielded ~60% recovery of protein at ~4.5 Fe/monomer as measured by a ferene assay as previously described [83], with protein quantified by Bradford protein assay with a correction of 0.69. Reconstituted protein was stored in a 4˚C refrigerator inside an anaerobic MBraun chamber.

**Determination of glycyl radical content of $St$NrdD.** Glycyl radical was installed on $St$NrdD via the activation reaction with $St$NrdG described below, then glycyl radical content was determined through EPR. Inside an anaerobic MBraun chamber, in a clear Eppendorf tube, 4 μM $St$NrdG monomer, 0.2 mM $S$-adenosylmethionine (AdoMet), 50 μM 5-deazaribo-flavin, 1% glycerol, and 10 μM $St$NrdD dimer were diluted and mixed in 30 mM potassium chloride, 30 mM bicine, pH 7.6. The $St$NrdD dimer was added last and, upon addition, the mixture was illuminated for 1 hr using a 2500 lumen LED bulb and the MBraun chamber light. After, the entire reaction mixture was transferred into an EPR tube, frozen in liquid nitrogen, and removed from the anaerobic chamber. Potassium nitrosodisulfonate (Fremy's salt) dissolved in anaerobic 30 mM potassium chloride, 30 mM bicine, pH 7.6 buffer were used as radical standards for EPR quantification of glycyl radical (S7 Fig). EPR (Bruker EMX-Plus) parameters were 80 K, 10 scans, modulation amplitude 3 G, center field 3350 G, sweep width 200 G, sweep time 21 s, microwave power 52 dB.

**Determination of protein melting temperature of $St$NrdD.** All data were collected on a LightCycler 480 (Roche) and analysis was done with LightCycler 480 Software release 1.5.0 SP3 (version 1.5.0.39). First, optimal ratios of $St$NrdD to SYPRO Orange (Sigma) dye were determined through performing the melting protocol while screening across 5X-20X SYPRO Orange and 0.5–10 μM $St$NrdD. SYPRO Orange and $St$NrdD stock solutions were diluted in 1X assay buffer (30 mM Tris pH 7.5, 30 mM potassium chloride, 10 mM magnesium sulfate), mixed in a white LightCycler 480 MultiWell Plate 96 plate (Roche), and sealed with a clear sealing film (Axygen). The melting protocol was as follows: the first target temperature was 20˚C, with a hold of 15 seconds and a ramp rate of 4.4˚C/s. The second target temperature was 99˚C, with a continuous acquisition mode of 10 acquisitions per ˚C and a ramp rate of 0.06˚C/s. The third target temperature was 20˚C, with a hold of 15 seconds and a ramp rate of 2.2˚C/s. Protein melting curves were analyzed within the LightCycler 480 software, using the $T_m$ Calling analysis to plot melting peaks (the negative first derivative of the melting curves). Melting temperatures were calculated through manual picking of the negative peaks of each run, within the LightCycler 480 software. The ratio of 4 μM $St$NrdD and 15X SYPRO Orange produced optimal curves. The melting temperature of $St$NrdD in the presence of nucleotides at assay concentrations was then determined. $St$NrdD, diluted in 1X assay buffer, was mixed with 1 mM GTP, 1 mM TTP and 3 mM ATP or 3 mM dATP and incubated at 37˚C for 2 min before addition of SYPRO Orange and further dilution with 1X assay buffer, supplemented with nucleotides to a ratio of 4 μM $St$NrdD and 15X SYPRO Orange. The same melting protocol and analysis was then performed as previously described.

### Activity assays

**Nucleotide preparation.** Nucleotides and deoxyribonucleotides (including heavy deoxyribonucleotides) were purchased from Santa Cruz Biotechnology or Sigma Aldrich. Nucleotides were dissolved to approximately 100 mM stock concentration in 50 mM HEPES and the pH was adjusted to approximately 8 before freezing in small aliquots for use in assays. Exact concentration was determined using UV/Vis spectroscopy and molar absorptivity values [84]. Heavy ($^{13}C/^{15}N$) nucleotides were shipped at 100 mM in 5 mM Tris buffer and were not diluted or buffer exchanged prior to use. All nucleotides were stored at -70˚C.

**Optimization of assay conditions.** To assay how quickly inactivation occurred (S1 Table), 40 µL of water was placed into a thin-walled PCR tube (USA Scientific) and was incubated for 5 minutes at 37˚C in a dry bath. The probe end of a multimeter set to the temperature setting was placed into the water and the tube was subsequently placed into a thermocycler set to 95˚C. Time to reach 45˚C, 60˚C, and 85˚C was measured for five aliquots and averaged. A control was also completed where the probe was directly put into a 95˚C solution of water and the time it took to reach 37˚C, 85˚C, and 95˚C was recorded. Note that because of the method of assaying temperature of the solution, the top of the thermocycler could not be closed during the temperature assay, and so heating may occur slightly more quickly during the assay.

Optimization of the dephosphorylation step (S2 Fig) was completed on solutions of 3 mM ATP and 1 mM CTP in class Ia assay buffer (50 mM HEPES pH 7.6, 15 mM magnesium chloride, 10 mM ethylenediaminetetraacetic acid). Standards were composed of 3 mM adenosine and 1 mM cytidine in class Ia assay buffer and were not treated with calf intestinal phosphatase. Varying volumes of calf intestinal phosphatase (Quick CIP; New England Biolabs) (0.5, 1, 3, or 5 µL) were added to 40 µL of solution and then were incubated in a thermocycler for 10 minutes, 30 minutes, 2 hours, or overnight (16 hours) prior to inactivation for 10 minutes at 85˚C. Samples were then filtered through a 0.2 µm filter (Costar) prior to analysis of A (m/z 268.2 -> 135.9) and C (m/z 243.2 -> 112.1) nucleosides via LC-MS/MS as described below. Optimization of the use of CutSmart buffer (New England Biolabs) was similarly performed on a solution of 40 µL of 3 mM ATP and 1 mM CTP in class Ia assay buffer. To this solution, 1 µL of CIP was added and either 4 µL of CutSmart buffer or 4 µL of assay buffer was added prior to dephosphorylation for 2 hours at 37˚C. The CIP was inactivated at 85˚C for 10 minutes prior to filtering in a 0.2 µm filter (Costar) and analysis of A and C nucleosides via LC-MS/MS as described below.

Optimization of standard curve conditions (S3 Fig) was completed using solutions containing 0–100 µM of either: (1) each of the four deoxyribonucleosides (dA, dC, dG, and dU), (2) each of the four deoxyribonucleotide products (dADP, dCDP, dGDP, and dUTP [used due to an unavailability of dUDP]), or (3) each of the four deoxyribonucleotide products in the presence of 3 mM ATP as an activity effector as well as the four substrates (100 µM ADP, 200 µM GDP, 50 µM UDP, 70 µM CDP). The three conditions were each completed in triplicate. The two conditions containing (deoxy)ribonucleotides were treated with 1 µL Quick CIP (New England Biolabs) for 2 hours at 37˚C prior to inactivation for 10 minutes at 85˚C and filtering through a 0.2 µm filter (Costar). The condition containing only deoxyribonucleosides was also filtered through a 0.2 µm filter, and then all three conditions were injected onto the LC-MS/MS for analysis of dA, dC, dG, and dU as described below.

**Aerobic assay.** *This assay was used to generate Figs 3, 4, 6–8.* All components of the assay (class Ia assay buffer, nucleotides, NrdA, Trx, TrxR, NADPH) minus NrdB were mixed in a reaction to generate 240 µL mastermix (final volume after NrdB addition) and incubated at 37˚C for two minutes. All proteins used in these assays have intact His tags, but it has been shown that His tags do not substantially interfere with NrdA and NrdB activity [74]. To ensure

that re-reduction was not the rate-limiting step, we used a large excess (60x) of Trx to ensure full and rapid reduction of NrdA. Nucleotide concentrations were generally modeled after physiological concentrations and are indicated below each Fig [71, 72]. After two minutes, a zero time point was generated by removing a volume of mastermix to a thin-wall PCR tube that would equal 40 μL after the addition of NrdB and was transferred to a thermocycler or heat block set at 95°C. After incubation for 2 minutes at 95°C to inactivate all proteins present, a volume of NrdB was added to the zero point to raise the volume to 40 μL and equal the final concentration of NrdB in the rest of the assay. The assay was then initiated by the addition of NrdB to the 37°C mastermix and 40 μL aliquots were taken into a thin-wall PCR tube at 30, 60, 90, and 120 s after initiation and immediately transferred to a 95°C thermocycler to stop the reaction. After all time points were taken, the time points were kept at 95°C for 2 more minutes and then cooled to 37°C for 10 minutes, at which point 1 μL calf alkaline intestinal phosphatase (Quick CIP; New England Biolabs) was added to each tube. The dephosphoryla-tion reaction was allowed to proceed for 2 hours at 37°C prior to inactivation of the CIP at 85°C for 2 minutes. For CDP reduction assays, samples containing internal standards (Fig 6), 1 μL of 1.6 mM heavy dCTP ($^{13}$C/$^{15}$N, Santa Cruz Biotechnology) was added to each 40 μL standard prior to treatment with CIP. The samples were then filtered through a 0.2 μm spin fil-ter plate (Pall) for 10 minutes at 1100 x g. Flow through was transferred to an LC-MS vial (Agi-lent Technologies) with a 150 μL insert prior to loading and running on the LC-MS/MS.

Standards consisted of 40 μL either dCDP (*E. coli* original assays; Figs 3, 4 and 6) or a mix-ture of dADP, dCDP, dGDP, and dUTP (*E. coli* 4 dN assays; Figs 7 and 8) in a 40 μL volume at concentrations of 0, 20, 40, 60, 80, and 100 μM of each nucleotide. All standards also contained 3 mM ATP. For each standard curve, standards were generated in triplicate, allowed to sit for 2 minutes at 37°C, and then transferred to 95°C for 2 minutes before being treated identically to inactivated samples from that point onwards (treatment with CIP, filtration, and transfer to vials). For standard curves containing internal standards (Fig 6), 1 μL of 1.6 mM heavy dCTP ($^{13}$C/$^{15}$N, Santa Cruz Biotechnology) was added to each 40 μL standard prior to treatment with CIP.

**Anaerobic assay.** *This assay was used to generate Fig 5.* In an anaerobic chamber (Mbraun) protein mastermix containing buffer (30 mM Tris pH 7.5, 30 mM potassium chlo-ride, 10 mM magnesium sulfate), DTT (5 mM), sodium formate (12.5 mM), and activated *St*NrdD (0.10 μM; still in mixture with *St*NrdG from the activation reaction) and a nucleotide mastermix containing all nucleotides needed (1 mM substrate GDP, 1 mM specificity effector TTP, 3 mM allosteric activator ATP [ATP condition] or 3 mM allosteric inhibitor dATP [dATP condition]) were prepared and mixed separately. All components were mixed to jointly generate a 250 μL combined mastermix. All components were gas exchanged to remove oxy-gen either by sparging with argon gas before bringing into the anaerobic chamber, or through passive gas exchange through venting in the anaerobic chamber for one hour up to overnight. The protein mastermix was incubated at 37°C for ten minutes and the nucleotide mastermix was incubated at 37°C for one minute. After incubation, a zero time point was generated by adding protein mastermix (40 μL minus the amount of nucleotide mastermix needed to add to equal the concentrations used in the rest of the assay) to a thin-walled PCR tube in a heat block at 95°C. After a two-minute incubation at 95°C to deactivate all proteins, a volume of nucleotide mastermix was added to the vial to raise the volume to 40 μL and equal the final concentration of nucleotides in the rest of the assay. The zero point was mixed and incubated for at least an additional two minutes. The assay was initiated by the addition of the nucleotide mastermix to the protein mastermix. At 30, 60, 90, and 120 seconds after initiation, 40 μL ali-quots were taken and immediately transferred to a thin-walled PCR tube in 95°C heat block to stop the reaction. The assay mixture was mixed before each aliquot was taken, and time points were timed so that the addition of assay mixture to the thin-wall PCR tube at 95°C signified

the end of a time point. After all time points were taken, the aliquots were kept at 95˚C for two more minutes to fully inactivate protein, at which point samples were removed from the anaerobic chamber. The samples were then treated with CIP, filtered, and transferred to LC-MS vials in an identical manner to the aerobic assay samples.

Standards consisted of 40 μL dGTP at concentrations of 0, 5, 10, 20, 30, 40, 60, 80, and 100 μM. All standards also contained 3 mM ATP, 1 mM GTP, 1 mM TTP, 5 mM DTT, 12.5 mM sodium formate in 30 mM Tris pH 7.5, 20 mM KCl, 10 mM MgSO$_4$. For each standard curve, standards were generated in triplicate, allowed to sit for 2 minutes at 37˚C, and then transferred to 95˚C for 2 minutes before being treated identically to inactivated samples from that point onwards (treatment with CIP, filtration, and transfer to vials).

**LC-MS/MS analysis.** A Synergi 2.5 μm Fusion-RP 100 Å LC column 100x2mm, particle size 2.5 (Phenomenex) with corresponding guard column was used for all experiments. LC-MS/MS was completed using an Agilent 6410 tandem triple quadrupole mass spectrometer connected to an Agilent 1100 HPLC system with 5 mM ammonium acetate pH 5.3 in water (buffer A) and acetonitrile (buffer B). LC-MS-grade ammonium acetate and acetic acid (Fisher Scientific) were used to generate buffer A, and buffer A was stored as a 1000x stock solution at -20˚C and diluted within one week of use. The flow rate was 0.350 mL/min throughout and separation was achieved over a 5 minute gradient from 3 to 5 percent B (S3 Table).

Column oven temperature was held at 35˚C throughout the runs. UV traces were monitored at 270 nm and desired analytes were monitored using MRM mode (fragmentor 165 V, dwell time 200 ms, collision energy 20 V). Analytes measured and their MRM parameters are indicated in S2 Table. Note that in cases where only one analyte was being measured, the program did not include the other analytes. Source parameters were gas temp 350˚C, gas flow 9 L/min, nebulizer 40 psi, capillary 4000 V (positive ion mode), and the electron multiplier voltage was set to 550 V. Injection volume was 10 μL per injection. All data processing and integration was completed using the Agilent MassHunter Qualitative Analysis program.

**Determination of protein activity from LC-MS/MS data.** The MRM peak for each analyte was integrated and the average +/- standard deviation integration of each standard was plotted against nmoles standard injected. A line of best fit was drawn through the standards and used to calculate the nmoles analyte(s) injected for each time point in the assay from the peak integrations. The nmoles analyte were graphed versus time in minutes and the slope of the line of best fit (in nmoles / minute) was calculated. The slope was divided by the total mg of the limiting protein present in each 10 μL injection to yield nmoles/mg-min for each time course. For aerobic assays, limiting protein was total mg of either NrdA or NrdB (not adjusted for radical content); for anaerobic assays, limiting protein was total mg activated NrdD present, with total mg NrdD adjusted based on concentration of glycyl radical identified in EPR analysis assuming maximal radical concentration of 2 radicals per dimer. The average activity and standard error of the mean of three time courses was taken for each condition. All graphical analysis was completed using Igor Pro v6.37.

## Supporting information

**S1 Table. Time taken for water to reach various temperatures upon insertion into a 95˚C thermocycler.** Each temperature was assayed five times using 40 μL of water in a thin-walled PCR tube that had an initial temperature of 37˚C.
(DOCX)

**S2 Table. MRM parameters for each of the four deoxyribonucleoside products and their corresponding internal standards quantified in the RNR reaction.**
(DOCX)

**S3 Table. LC-MS program used for the LC-MS/MS experiments.**
(DOCX)

**S1 Fig. Protein melting peaks of *St*NrdD in different nucleotide conditions.** Protein melting peaks of *St*NrdD with no nucleotides (*St*NrdD alone) or 1 mM TTP and GTP and either 3 mM ATP or 3 mM dATP. Table insert summarizes the melting temperature of *St*NrdD under the three conditions, which equates to the temperature at the negative peak of each curve.
(TIF)

**S2 Fig. Optimization of dephosphorylation step using calf intestinal phosphatase (CIP).** **(A)** Variation in dephosphorylation based on volume CIP added and incubation time (at 37°C) on a solution of 3 mM ATP and 1 mM CTP. Dephosphorylation was determined by adenosine MRM signal and was compared to a standard of 3 mM adenosine nucleoside. 1 μL CIP addition and incubation for 2 hours was used for all further studies. **(B)** Impact of the addition of CutSmart buffer to the CIP incubation mixture on a solution of 3 mM ATP and 1 mM CTP. Dephosphorylation was determined by cytidine MRM signal and was compared to a standard of 1 mM cytidine nucleoside. No CutSmart buffer was added in any further studies.
(TIF)

**S3 Fig. Standard curve optimization.** The presence of substrate and effector nucleosides and/ or phosphates in addition to the measured deoxyribonucleoside standard is essential in generation of accurate standard curves. A comparison of dA, dC, dG, and dU standard curves generated either from deoxyribonucleotide triphosphates in the presence of substrates and effectors (navy), deoxyribonucleotide triphosphates but no substrates or effectors (purple), or only deoxyribonucleosides (no phosphates; light blue). For conditions where nucleotides were present, nucleotides were treated with CIP prior to analysis by mass spectrometry. Concentrations of substrate and effector nucleotides were as follows: ADP (110 μM), GDP (200 μM), UDP (50 μM), CDP (70 μM), ATP (3 mM). All products measured by LC-MS/MS ranged from 0–100 μM.
(TIF)

**S4 Fig. TIC and MRM curves for the four ribonucleotide substrate experiment depicted in Fig 7.** **(A)** TIC and MRM curves for each of the four standard curves. **(B)** TIC and MRM curves for the activity assay. The above condition contained Trx (30 μM), TrxR (0.5 μM), NADPH (200 μM), ATP (3000 μM), dATP (500 μM), TTP (250 μM), dGTP (100 μM), CDP (70 μM), UDP (50 μM), GDP (200 μM), ADP (110 μM), NrdA (0.1 μM dimer), and NrdB (0.5 μM dimer).
(TIF)

**S5 Fig. Limits of detection of the four deoxynucleosides on the Agilent 6410 system. (A)** Lower limit of detection for each of the four deoxynucleosides. **(B)** Standard curves are linear at up to 10 nmoles injected deoxyribonucleoside for each product. Note that the upper limit of detection and the upper linear range are both likely higher but are not anticipated to be relevant to RNR assays. Each data point is the mean ± standard error of the mean of three replicates.
(TIF)

**S6 Fig. EPR spectra of native (green) and reconstituted (blue) NrdB.** EPR parameters were 80K, 10 scans, center field 3330 G, sweep width 300 G, sweep time 20 s, microwave power 38dB, modulation amplitude 1.5G. Double integrals were 134.5 and 580.6 for 100 μM native and reconstituted protein, respectively, leading to radical content of 0.3 and 1.2 radicals per

dimer, respectively.
(TIF)

**S7 Fig. EPR spectra of the glycyl radical in wild-type *St*NrdD.** EPR parameters were 80K, 10 scans, modulation amplitude 3G, center field 3350 G, sweep width 200G, sweep time 21 s, microwave power 52dB.
(TIF)

## Acknowledgments

The authors gratefully acknowledge the support of Dr. Michael DeMott at the Center for Environmental Health Sciences for valuable discussions and troubleshooting of the LC-MS/MS assay and instrumentation, and Michael A. Funk for cloning *S. thermophilus* NrdD. The authors additionally thank Dr. Mary Andorfer, Dr. Kelsey Miller, and Alison Biester for their assistance with collection of EPR spectra.

## Author Contributions

**Conceptualization:** Talya S. Levitz, Rohan Jonnalagadda, Christopher D. Dawson.

**Formal analysis:** Talya S. Levitz, Gisele A. Andree.

**Funding acquisition:** Catherine L. Drennan.

**Investigation:** Talya S. Levitz, Gisele A. Andree.

**Methodology:** Talya S. Levitz, Gisele A. Andree, Rohan Jonnalagadda, Christopher D. Dawson, Rebekah E. Bjork.

**Supervision:** Catherine L. Drennan.

**Validation:** Talya S. Levitz, Gisele A. Andree.

**Writing – original draft:** Talya S. Levitz, Gisele A. Andree, Catherine L. Drennan.

**Writing – review & editing:** Talya S. Levitz, Gisele A. Andree, Rohan Jonnalagadda, Christopher D. Dawson, Rebekah E. Bjork, Catherine L. Drennan.

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
