## [Decision Letter · Decision Letter 0]

15 Mar 2022

PONE-D-22-03785A rapid and sensitive assay for quantifying the activity of both aerobic and anaerobic ribonucleotide reductases acting upon any or all substratesPLOS ONE

Dear Prof Drennan,

Thank you for submitting your manuscript to PLOS ONE. After careful consideration, we feel that it both has merit and would benefit from addressing the comments of the reviewers to improve the manuscript prior to publication. Therefore, we invite you to submit a revised version of the manuscript with these points raised during the review process addressed. Specifically, reviewer 3 presents a number of points that we feel would help to clarify the manuscript and support wider uptake of the protocols outlined, alongside improvements suggested by the other reviewers (please see the attachments for details).

We look forward to receiving your revised manuscript.

Kind regards,

Anna Kristina Croft

Academic Editor

PLOS ONE

Journal Requirements:

Reviewers' comments:

Reviewer's Responses to Questions

**Comments to the Author**

1. Is the manuscript technically sound, and do the data support the conclusions?

Reviewer #1: Yes

Reviewer #2: Yes

Reviewer #3: Partly

2. Has the statistical analysis been performed appropriately and rigorously? 

Reviewer #1: Yes

Reviewer #2: I Don't Know

Reviewer #3: Yes

3. Have the authors made all data underlying the findings in their manuscript fully available?

Reviewer #1: Yes

Reviewer #2: Yes

Reviewer #3: Yes

4. Is the manuscript presented in an intelligible fashion and written in standard English?

Reviewer #1: Yes

Reviewer #2: Yes

Reviewer #3: No

5. Review Comments to the Author

Reviewer #1: Summary: Drennan et al. present a versatile assay for RNR activity. This assay has several benefits: it does not require radiolabeled nucleotide substrates and the associated infrastructure for working with and quantitating radioactive samples, it enables rapid and semi-high throughput product quantitation which in turn allows a wide variety of conditions to be screened, and it is applicable to a variety of RNRs from different classes and organisms. The conclusions are well-supported and are of broad interest to the readership of PLOS ONE. Suggestions to improve the clarity of the presented work are attached, but with these emendations, I strongly support publication. The initial assay results reported already reveal important new insights into RNR allostery.

Reviewer #2: The presented manuscript describes the development of an LC-MS/MS activity assay for the partial characterisation of ribonucleotide reductases (RNs) by making use of their intrinsic allosteric inhibition which determines the product distribution. Thus, this method can assay RNs conversion of ribonucleotides (substrates) in the presence of all four at the same time. Further, the method is faster and does not require handling of radioactive materials.

Overall, the manuscript is very well written and interesting to read. I am not an expert in the field but had overall no difficulties understanding it.

My only suggestion is a linguistic one – the meaning in line 347, I had to read a couple of times until I got the pieces together. Would it be possible changing “prior to use of heat” to “before using/applying heat”?

Recommend for publication as it stands.

Reviewer #3: Answers to questions above:

1. There are some assumptions and activity adjustments in the manuscript which are not specified.

4. There are more than one incomplete sentences.

Review uploaded as an attachment

6. PLOS authors have the option to publish the peer review history of their article (what does this mean?). If published, this will include your full peer review and any attached files.

Reviewer #1: No

Reviewer #2: **Yes: **Martina Lahmann

Reviewer #3: No

---

## [Author Response · Author response to Decision Letter 0]

28 Apr 2022

See attached point-by-point response letter

---

## [Decision Letter · Decision Letter 1]

19 May 2022

PONE-D-22-03785R1A rapid and sensitive assay for quantifying the activity of both aerobic and anaerobic ribonucleotide reductases acting upon any or all substratesPLOS ONE

Dear Cathy,

Only one minor point to address from the reviewer; please see review comments. 

We look forward to receiving your revised manuscript.

Kind regards,

Anna Kristina Croft

Academic Editor

PLOS ONE

Journal Requirements:

Reviewers' comments:

Reviewer's Responses to Questions

**Comments to the Author**

1. If the authors have adequately addressed your comments raised in a previous round of review and you feel that this manuscript is now acceptable for publication, you may indicate that here to bypass the “Comments to the Author” section, enter your conflict of interest statement in the “Confidential to Editor” section, and submit your "Accept" recommendation.

Reviewer #3: (No Response)

2. Is the manuscript technically sound, and do the data support the conclusions?

Reviewer #3: Yes

3. Has the statistical analysis been performed appropriately and rigorously? 

Reviewer #3: Yes

4. Have the authors made all data underlying the findings in their manuscript fully available?

Reviewer #3: Yes

5. Is the manuscript presented in an intelligible fashion and written in standard English?

Reviewer #3: Yes

6. Review Comments to the Author

Reviewer #3: The authors have responded satisfactorily to all questions, except for one point. It is still not clear from the determination of protein activity from LC-MS/MS data how the activity of E. coli NrdB is calculated. The radical content of EcNrdB was determined to 1.2 radicals per dimer, and the authors state in their response that “we believe that NrdB must have 2 radicals/dimer to be active”. Is the activity of 10 000 nmol dCDP/mg-min for EcNrdB adjusted for radical content or not? The question is valid, as the activity of S. thermophilus NrdD is adjusted for radical content. Whether the activity of EcNrdB has been adjusted for radical content or not needs to be clearly spelled out in the section on lines 252-266 where EcNrdB activity is reported or in the section on lines 855-864 where the adjustment of StNrdD activity is described.

7. PLOS authors have the option to publish the peer review history of their article (what does this mean?). If published, this will include your full peer review and any attached files.

Reviewer #3: No

---

## [Author Response · Author response to Decision Letter 1]

22 May 2022

Response to one remaining reviewer comment

Reviewer #3: The authors have responded satisfactorily to all questions, except for one point. It is still not clear from the determination of protein activity from LC-MS/MS data how the activity of E. coli NrdB is calculated. The radical content of EcNrdB was determined to 1.2 radicals per dimer, and the authors state in their response that “we believe that NrdB must have 2 radicals/dimer to be active”. Is the activity of 10 000 nmol dCDP/mg-min for EcNrdB adjusted for radical content or not? The question is valid, as the activity of S. thermophilus NrdD is adjusted for radical content. Whether the activity of EcNrdB has been adjusted for radical content or not needs to be clearly spelled out in the section on lines 252-266 where EcNrdB activity is reported or in the section on lines 855-864 where the adjustment of StNrdD activity is described.

The activity of 10 000 nmol dCDP / mg-min for EcNrdB is not adjusted for radical content. We had added a sentence describing this in the same area of the methods where the adjustment is described for NrdD:

The slope was divided by the total mg of the limiting protein present in each 10 µL injection to yield nmoles/mg-min for each time course. For aerobic assays, limiting protein was total mg of either NrdA or NrdB (not adjusted for radical content); for anaerobic assays, limiting protein was total mg activated NrdD present, with total mg NrdD adjusted based on concentration of glycyl radical identified in EPR analysis assuming maximal radical concentration of 2 radicals per dimer.

---

## [Editor Report · Decision Letter 2]

24 May 2022

A rapid and sensitive assay for quantifying the activity of both aerobic and anaerobic ribonucleotide reductases acting upon any or all substrates

PONE-D-22-03785R2

Dear Prof. Drennan,

We’re pleased to inform you that your manuscript has been judged scientifically suitable for publication and will be formally accepted for publication once it meets all outstanding technical requirements.

Kind regards,

Anna Kristina Croft

Academic Editor

PLOS ONE
---

## [Editor Report · Acceptance letter]

30 May 2022

PONE-D-22-03785R2 

A rapid and sensitive assay for quantifying the activity of both aerobic and anaerobic ribonucleotide reductases acting upon any or all substrates 

Dear Dr. Drennan:

I'm pleased to inform you that your manuscript has been deemed suitable for publication in PLOS ONE. Congratulations! Your manuscript is now with our production department. 

Kind regards, 

on behalf of

Dr. Anna Kristina Croft 

Academic Editor

PLOS ONE